# Identification and functional characterization of muscle satellite cells in *Drosophila*

**Dhananjay Chaturvedi[1†], Heinrich Reichert[2], Rajesh D Gunage[1†‡*], K VijayRaghavan[1*]**

[1]Department of Developmental Biology and Genetics, National Center for Biological Sciences, Tata Institute of Fundamental Research, Bangalore, India; [2]The Centre for Molecular Life Sciences, Biozentrum, Basel, Switzerland

**Abstract** Work on genetic model systems such as *Drosophila* and mouse has shown that the fundamental mechanisms of myogenesis are remarkably similar in vertebrates and invertebrates. Strikingly, however, satellite cells, the adult muscle stem cells that are essential for the regeneration of damaged muscles in vertebrates, have not been reported in invertebrates. In this study, we show that lineal descendants of muscle stem cells are present in adult muscle of *Drosophila* as small, unfused cells observed at the surface and in close proximity to the mature muscle fibers. Normally quiescent, following muscle fiber injury, we show that these cells express Zfh1 and engage in Notch-Delta-dependent proliferative activity and generate lineal descendant populations, which fuse with the injured muscle fiber. In view of strikingly similar morphological and functional features, we consider these novel cells to be the *Drosophila* equivalent of vertebrate muscle satellite cells.
DOI: https://doi.org/10.7554/eLife.30107.001

**\*For correspondence:**
rajeshgunage@gmail.com (RDG);
vijay@ncbs.res.in (KVR)

[†]These authors contributed equally to this work

Present address: [‡]Stem Cell Program and Division of Haematology/Oncology, Children's Hospital, Howard Hughes Medical Institute, Boston, United States

## Introduction

A great deal of insight into the cellular and molecular mechanisms of muscle development has been obtained in two powerful genetic model systems, the mouse and *Drosophila.* Despite numerous differences in the specific ways in which muscles are formed in these two organisms, there are also remarkable similarities in the fundamental developmental processes that underlie myogenesis (*Roy and VijayRaghavan, 1999*; *Rai et al., 2014*; *Bothe and Baylies, 2016*; *Schnorrer et al., 2010*). These similarities are most clearly evident when the mechanisms of myogenesis of the large multifibrillar indirect flight muscles of *Drosophila* are compared to vertebrate skeletal muscles. In both cases, muscle stem cells generated during embryogenesis give rise to a large pool of muscle precursor cells called myoblasts that subsequently fuse and differentiate to produce the multinucleated syncytial cells of the mature muscle. These mechanistic similarities of myogenesis are reflected at the molecular genetic level, in that many of the key genes involved in *Drosophila* muscle development have served as a basis for the identification of comparable genes in vertebrate muscle development (e.g. (*Srinivas et al., 2007*; *Abmayr and Pavlath, 2012*).

In vertebrates, mature skeletal muscle cells can manifest regenerative responses to insults due to injury or degenerative disease. These regenerative events require the action of a small population of tissue specific stem cells referred to as satellite cells (*Bothe and Baylies, 2016*; *Mauro, 1961*; *Brack and Rando, 2012*; *Relaix and Zammit, 2012*). Muscle satellite cells are located between the sarcolemma and the basal lamina of muscle fibers. Although normally quiescent, satellite cells respond to muscle damage by proliferating and producing myoblasts, which differentiate and fuse with the injured muscle cells. Myoblasts generated by satellite cells are also involved in the growth

of adult vertebrate muscle. Given the numerous fundamental aspects of muscle stem cell biology and myogenesis that are similar in flies and vertebrates, it is surprising that muscle satellite cells have not been reported in *Drosophila*. Indeed, due to the apparent absence of satellite cells in adult fly muscles, it is unclear if muscle regeneration in response to injury can take place in *Drosophila*.

In a previous study, we showed that a small set of embryonically generated muscle-specific stem cells known as AMPs (adult muscle progenitors) give rise post-embryonically to the numerous myoblasts which fuse to form the indirect flight muscles of adult *Drosophila* (*Gunage et al., 2014*). Here, we show that muscle stem cell lineal descendants are present in the adult as unfused cells which have all the anatomical features of muscle satellite cells. In adult muscle, these unfused cells are located in close proximity to mature muscle fibers and are surrounded by the basal lamina of the fibers. Moreover, although normally quiescent, following muscle injury they undergo Notch signaling-dependent proliferation to generate fusion-competent lineal descendants. In view of these remarkable developmental, morphological and functional features, we consider these cells to be the *Drosophila* equivalent of vertebrate muscle satellite cells. Thus, in flies and vertebrates the muscle stem cell lineage that generates the adult-specific muscles during normal development is also available for adult myogenesis in muscle tissue in response to damage. This finding further opens adult *Drosophila* muscle for the understanding of muscle maintenance, wasting, damage and repair.

## Results

### Two different types of cells are present in adult flight muscle

During normal postembryonic development of the indirect flight muscles, a set of approximately 250 mitotically active adult muscle precursors (AMPs) located on the epithelial surface of the wing imaginal disc generates a large number of postmitotic myoblast progeny. These myoblasts subsequently migrate and fuse to form the indirect flight muscles (IFMs) of the adult. The IFMs are composed of the dorso-ventral muscles (DVMs) formed by the de novo fusion of myoblast and the dorsal longitudinal muscles (DLMs) which are formed using remnant larval muscles as templates (*Gunage et al., 2014*; *Fernandes et al., 1991*; *Dhanyasi et al., 2015*).

Consistent with their developmental origin, which results in a large myoblast pool, the IFMs are large multinucleated cells formed by myoblast fusion. For convenience we focus here on the DLMs. The multinucleate nature of these muscles is evident in confocal optical sections through adult flight muscle fibers labeled by TOPRO (marks all nuclei) and myosin heavy chain (MHC) immunostaining, which marks myofibers. As expected, numerous nuclei, clearly located intra-cellular between individual myofibrils, are seen throughout the muscle fiber (*Figure 1A, A'*, white arrowheads). Interestingly, however, these optical sections also reveal nuclei located peripherally in close proximity to the muscle fiber surface (*Figure 1A, A'*, green arrowheads).

Additional co-labeling of these adult muscle fibers with Dmef2-Gal4 driving mCD8GFP (marking muscle fiber membranes) indicates that these peripherally located nuclei belong to cells at the muscle fiber surface, which are apparently not fused with their associated muscle fiber cell (*Figure 1B, B'*, white arrowheads). *Figure 1C, D* show some these nuclei at higher magnification revealing that they are located at the muscle fiber surface and are surrounded by membrane-specific mCD8GFP label implying that these nuclei are not situated inside the muscle fiber. This observation is confirmed by co-staining these adult muscle fibers for expression of either Act88F, an indirect flight muscle-specific isoform of actin, or Tropomyosin (*Figure 1—figure supplement 1*).

Scans along the z-axis through co-labeled optical sections of muscle fibers indicate a small number of GFP-positive cells located closely associated with the surface muscle of fibers but remain unfused. To determine the relative numbers of peripherally located unfused cells versus fused differentiated myoblasts, all optical sections (along the z-axis) of the co-labeled adult DLM muscle fibers were scored for cells associated with the surface of the muscle fibers versus cell nuclei located within the flight muscle fibers. These experiments (n = 12) indicate that a DLM muscle fiber has an average of 20 ± 4 unfused cells versus an average of 700 ± 50 fused myoblasts. Hence the ratio of unfused cells to differentiated fused myoblasts is 1:30 implying that the population of surface-associated cells is markedly smaller than the population of fused myoblasts.

Taken together, these findings indicate that two different types of cells are present in adult muscle. The first comprises the well-characterized population of differentiated myoblasts that have fused

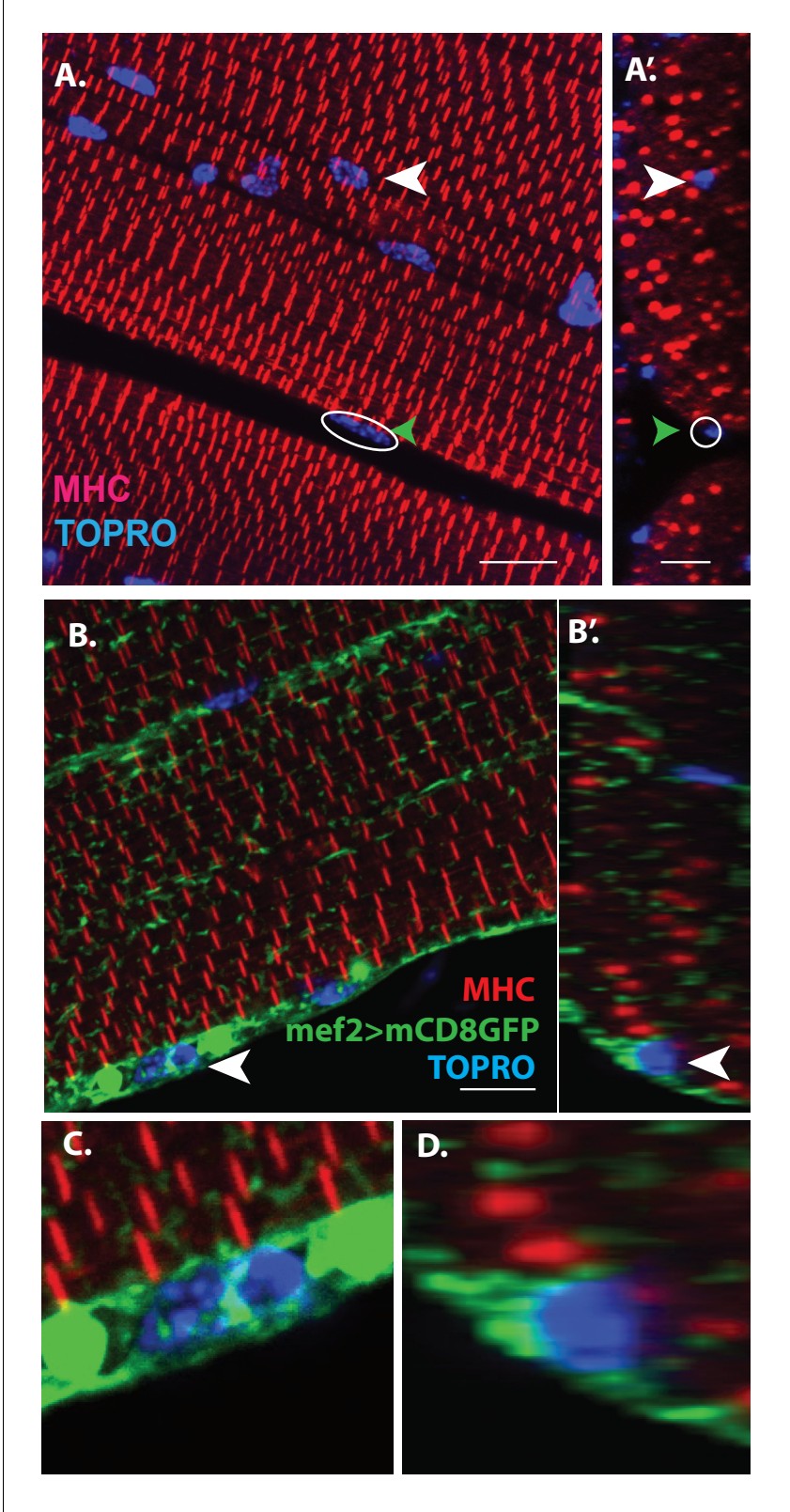

**Figure 1.** Unfused muscle associated cells are present at the surface of adult flight muscles. (**A**) Dorsal longitudinal muscles (DLMs) stained for Myosin Heavy Chain (MHC) (red) to delineate muscle fibrils and TOPRO (Blue) marking nuclei. White arrowhead marks one example nucleus, surrounded by MHC-labeled myofibrils showing it is inside a myofiber. Green arrowhead, white circle, marks one example nucleus located at the peripheral surface of MHC

*Figure 1 continued on next page*

*Figure 1 continued*
labeled myofiber. (**A'**) Orthogonal view of (**A**). (**B**) DLMs stained for Dmef2 Gal4 > UAS mCD8::GFP marking muscle membrane (green) MHC (red) and TOPRO (blue). Unfused nuclei are enveloped in GFP-labeled membrane (white arrowhead), **B'** Orthogonal view of (**B**) at the muscle fiber surface. (**C, D**) Magnified views of nucleus indicated in (**A'**) and (**B'**), respectively. N = 15. Scale bar 10 μm.
DOI: https://doi.org/10.7554/eLife.30107.002
The following figure supplement is available for figure 1:

**Figure supplement 1.** Unfused muscle-associated cells at the surface of adult flight muscles are excluded by Act88F and Tropomyosin inside myofibers.
DOI: https://doi.org/10.7554/eLife.30107.003

to generate the large multinuclear muscle fibers. The second comprises a novel population of small, apparently unfused cells located at the surface of the muscle fibers. In the following, we will refer to these small, unfused muscle fiber-associated cells as *Drosophila* satellite cells.

## Ultrastructure of satellite cells in adult flight muscle

To characterize the morphological features of the close association of satellite cells with the large multinucleated muscle fibers at the ultrastructural level, an electron-microscopic analysis of adult DLM fibers was carried out. In electron micrographs, the mature muscle fibers are large cells containing multiple prominent nuclei, numerous organelles, as well as extensive sets of elongated myofibrils, and are surrounded by a prominent extracellular matrix (*Figure 2A*). In addition to these typical multinucleated muscle cells, the ultrastructural analysis also shows satellite cells as small, wedge-shaped cells closely apposed to the large multinucleated muscle fibers (*Figure 2B*, *Figure 2—figure supplement 1*). These satellite cells have compact nuclei and small cytoplasmic domains with few organelles. The intact cell membrane of the satellite cells is directly adjacent to the intact muscle cell membrane demonstrating unequivocally that they are not fused with the muscle cells. They do, however, appear to be embedded in the same contiguous extracellular matrix of their adjoining muscle fiber.

In terms of their ultrastructural morphology, the satellite cells in adult flight muscle share significant characteristics with satellite cells of vertebrate muscle. In both cases, the cells are small, mononucleated and intercalated between the cell membrane and the extracellular matrix of mature muscle fibers.

## *Drosophila* satellite cells are lineal descendants of adult muscle precursors

Previous work has shown that the myoblasts which fuse to generate adult muscle derive from a small set of stem cell-like AMPs (*Gunage et al., 2014*). Proliferating AMPs located on the larval wing disc can be identified by clonal MARCM labeling experiments using a Dmef2-Gal4 driver (*Lee and Luo, 2001*; *Wu and Luo, 2006*); (*Figure 3A*).

Given the fact that *Drosophila* satellite cells, like myoblasts, are labeled by the Dmef2-Gal4 driver and considering their close ultrastructural association with muscle fibers, we wondered if these satellite cells might also be lineal descendants of AMPs (*Figure 3B*). To investigate this, we induced MARCM clones in late larval stages and recovered labeled clones in the adult muscle. In these experiments, Dmef2-Gal4 was used to drive a GFP reporter label, muscle cells were co-labeled using MHC immunostaining, and cell nuclei were co-labeled with TOPRO. Labeled cells in the adult were visualized using confocal microscopy and analyzed in serial stacks of optical sections.

These clonal labeling experiments reveal the presence of a small number of GFP labeled satellite cell nuclei closely apposed to the surface of the adult muscle fibers. A reconstructed 3D view of optical sections shows that these GFP-positive nuclei are distributed along the entire surface of the muscle fibers and located both at the interface between different muscle fibers and at the surface of individual muscle fibers but not within the muscle fibers (*Figure 3C,D*). Note that while the nuclei of differentiated myoblasts within the muscle fiber are also targeted in this MARCM experiment, no clonal UAS-GFP labeling is visible due to the persistent expression of the Gal80 repressor by the unlabeled nuclei in the multinuclear muscle cell. Hence, in these experiments, cell nuclei can only be labeled if their cells are unfused and remain outside of the muscle fiber.

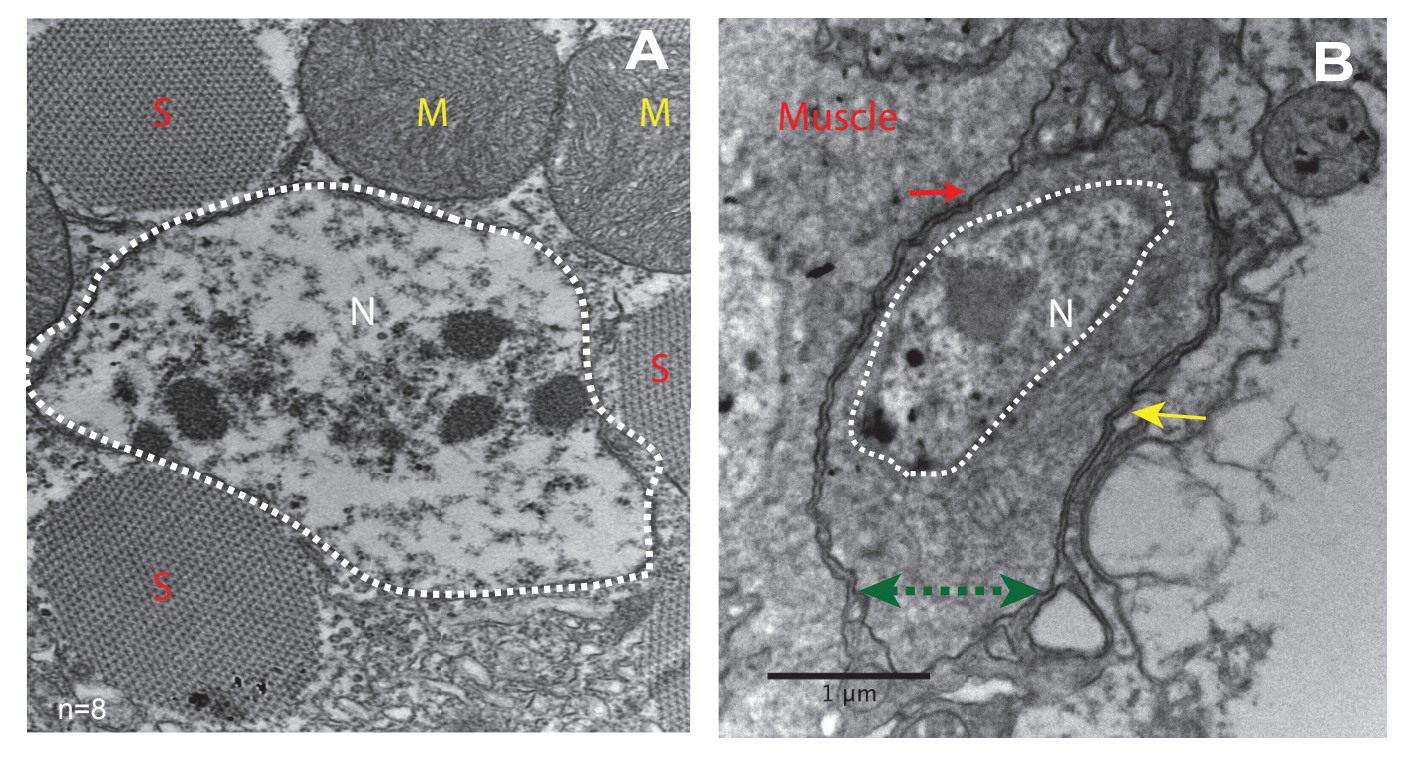

**Figure 2.** Unfused muscle-associated cells have ultrastructural features of satellite cells. (A, B) Transmission electron micrographs of adult flight muscle. (A) Nuclei inside DLM fibers are large round structures surrounded by nuclear membranes (white dotted lines). A. Distinct sarcomeres in the cytoplasm of the muscle syncytium (marked as S) and mitochondria (marked as M). (B) Mononucleate cell apposed to mature muscle surface. Cell membrane (marked by a green double-headed arrow) seen distinctly apposed to mature muscle membrane (red arrow) and beneath the basement membrane (yellow arrow) of the muscle fiber. Organelles and wedged shaped nucleus (white dotted line) are visible in the cytoplasm of this cell. N = 8 Scale bar 1 μm.

DOI: https://doi.org/10.7554/eLife.30107.004

The following figure supplement is available for figure 2:

**Figure supplement 1.** Electron micrograph of unfused mononucleate cell at muscle surface.

DOI: https://doi.org/10.7554/eLife.30107.005

These clonal MARCM labeling experiments are in accordance with the notion that Drosophila satellite cells are lineal descendants of AMPs, which, in contrast to myoblasts, do not fuse with the mature muscle cells but rather persist as unfused cells in the adult albeit closely associated with the mature muscle fibers. To control for a possible contribution of hemocytes lineages to the unfused cell population, we used the e33c-Gal4 line to show that hemocytes are not seen in muscles (*Figure 3—figure supplement 1*; for expression pattern of the e33c-Gal4 in hemocytes see *Fossett et al. (2003)* and *Matova and Anderson (2006)*.

## The Zfh1 transcription factor is a marker for satellite cells in adult muscle

To facilitate the analysis of satellite cells, we searched for a protein specifically expressed in adult DLM satellite cells by examining the expression of a set of transcription factors implicated in embryonic muscle specification and myoblast fusion competence. Among these, we identified Zfh1 as a specific marker for adult muscle satellite cells. Zfh1is a zinc finger transcription factor that regulates somatic myogenesis from embryonic stages onward (*Postigo and Dean, 1999*; *Sellin et al., 2009*). In larval stages, Zfh1 is expressed in all of the AMP lineage cells on the wing discs; both AMPs and myoblasts are labeled (*Figure 4A*).

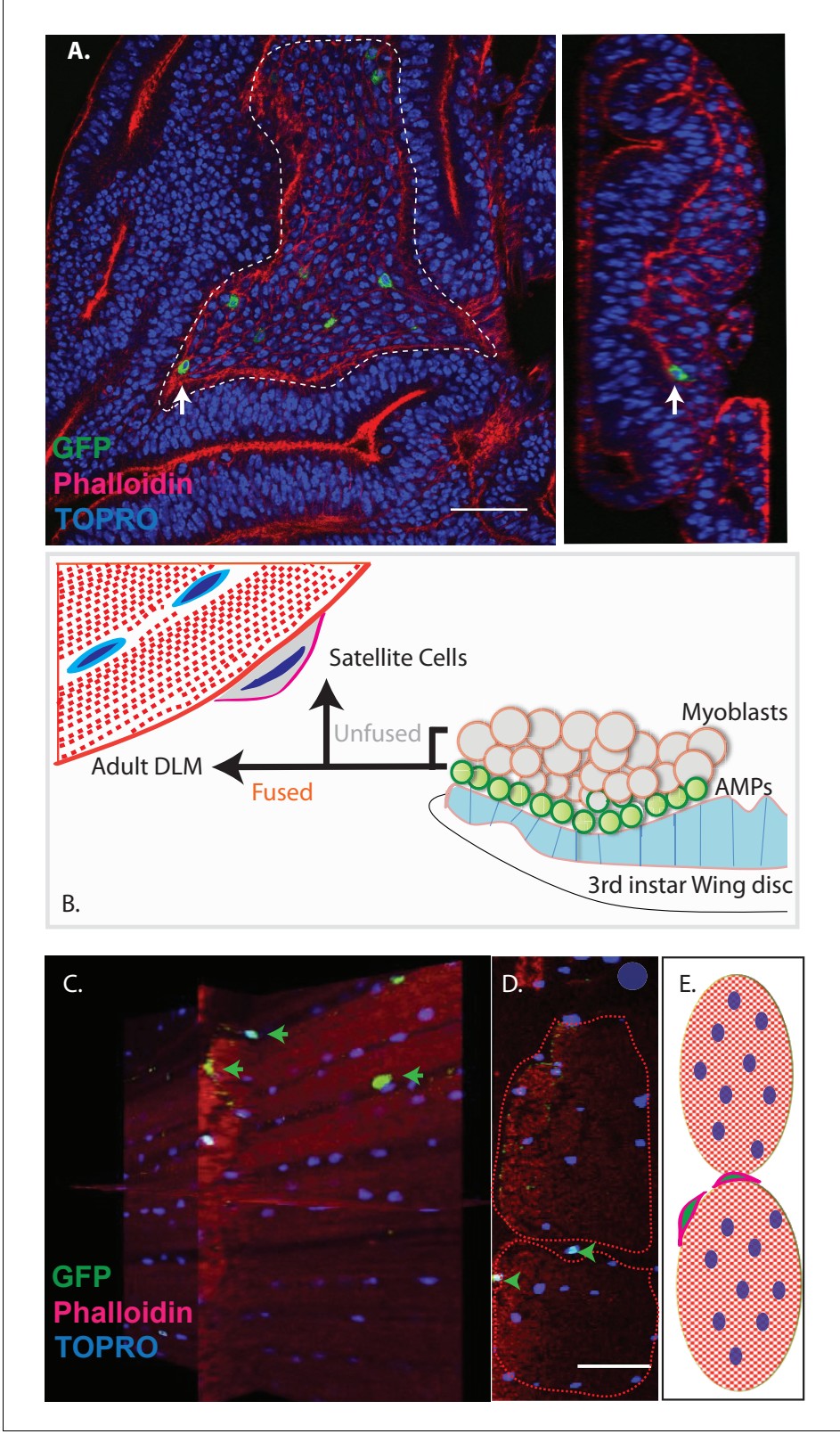

**Figure 3.** Unfused cells of the AMP lineage persist in adult muscle. (A) Single-cell MARCM clones of AMP lineage (*mef2*-Gal4 driver) induced in the third instar (120AEL) and recovered from a single 15 m heat shock at 37°C, clones from the notum of the wing disc (induced in the late third instar (120AEL). A labeled single cell clone (green) is indicated by a white arrow. Right panel shows the same cell (white arrow) in orthogonal view. The AMP

*Figure 3 continued on next page*

*Figure 3 continued*

cell lies in close proximity to wing disc epithelium. Phalloidin (red) marks F-actin and TOPRO (blue) marks all the nuclei. (B). Simplified schematic describing lineal origin of adult DLM fibers and satellite cells. AMPs (green circles) on the third instar wing disc notum give rise to myoblasts (beige circles) located distal to the epithelium. Cells from the AMP lineage either fuse to muscle templates and give rise to adult DLMs or remain unfused as mononucleate cells closely apposed to the DLM surface. (C). MARCM clones with *mef2*-Gal4 driving UAS mCD8:: GFP induced in the third instar (~120 hr AEL) and recovered in the adults stage. 3D reconstruction of adult muscle with mononucleate GFP-labeled cells (green arrows) located on the surface of mature DLM fibers. Phalloidin (red) marks F-actin and TOPRO (blue) marks all the nuclei. (D) Orthogonal view of the same preparation as in (C). *mef2*-Gal4-labeled mononucleate MARCM clones (GFP positive) indicated with green arrowheads clearly seen on the surface of adult DLMs. (E) Simplified schematic of (D) Red checkered ovals containing blue ovals indicate mature DLMs. Cells with red membranes and green nuclei represent satellite cells. Scale bars in (A, C, D) 50 μm. N = 12.
DOI: https://doi.org/10.7554/eLife.30107.006

The following figure supplement is available for figure 3:

**Figure supplement 1.** DLMs showing absence of hemocytes.
DOI: https://doi.org/10.7554/eLife.30107.007

In contrast, in adults, Zfh1 is specifically and exclusively expressed in the unfused muscle satellite cells. Thus, in adult muscle, cells labeled by a Zfh1 antibody are clearly unfused and located closely apposed to the mature muscle fibers (*Figure 4B–D*). Similar highly specific labeling of unfused muscle satellite cells is observed in adult muscle with either a GFP-tagged Zfh1 protein or with a Zfh1-fused Gal4 driver and a UAS-RedStinger reporter (*Figure 4E,F*) (*Puretskaia et al., 2017*). In all cases, Zfh1 expression is limited to unfused cells and is never seen inside intact adult muscle fibers.

Since Zfh1 is expressed in all AMP lineage cells in larval stages but is restricted to unfused satellite cells in the adult, its expression pattern must change dramatically in pupal stages. To document this, we carried out an analysis of Zfh1 expression at different time points during DLM development in representative pupal stages.

Indirect flight muscle development during pupal stages has been characterized in detail (*Fernandes et al., 1991*; *Roy and VijayRaghavan, 1998*; *Bate et al., 1991*). During the earliest pupal stages at the onset of metamorphosis, the AMP-descendent myoblasts are still located on the third instar wing disc notum. Subsequently, these myoblasts migrate from the wing disc and begin to swarm over a set of persistent larval muscles referred to as DLM templates that act as scaffolds for the developing DLMs. By 20 hr after puparium formation (APF) numerous myoblasts are present around and between the six DLM templates, fusion of myoblasts with these transformed DLM templates is ongoing, and myoblast nuclei are observed inside the developing DLMs. By 30 hr APF, most myoblasts have fused with the DLMs and by 36 hr APF, myogenesis of the adult DLM muscle is essentially complete.

Immunolabeling experiments show that at 20 hr APF all of the nuclei of myoblasts around and in between the DLM templates express Zfh1 (*Figures 5A* and *6A*). Moreover, the nuclei that are located inside the DLM templates due to the fusion of their myoblasts with the templates also express Zfh1. Thus, at this pupal stage, most if not all the nuclei of the AMP lineage myoblasts, be they fused within DLM templates or unfused outside of these templates, continue to express Zfh1 comparable to the situation in larval stages (see *Figure 4A*). In contrast, by 30 hr APF a dramatic change in the number and location of Zfh1 expressing cell nuclei has occurred. Zfh1 expression is only seen in a very small number of unfused cell nuclei located outside, albeit closely apposed to, the DLM muscle fiber templates (*Figures 5B* and *6B*). None of the nuclei located inside of the developing DLM muscle fibers express Zfh1. Thus, by 30 hr APF, when myoblast fusion is largely complete and myofibrils become visible in the muscle, Zfh1 expression has become restricted to a small number of unfused AMP lineal cells comparable to the situation in the adult, in which the only AMP lineal descendants that continue to express Zfh1 are the unfused satellite cells.

Taken together, these findings establish Zfh1 expression as a specific marker for satellite cells in adult muscle. Moreover, they characterize the dramatic change in Zfh1 expression that occurs in DLM muscle development between 20 hr and 30 hr APF, and provide further support for the lineal origin of satellite cells from the muscle stem cell-like AMPs.

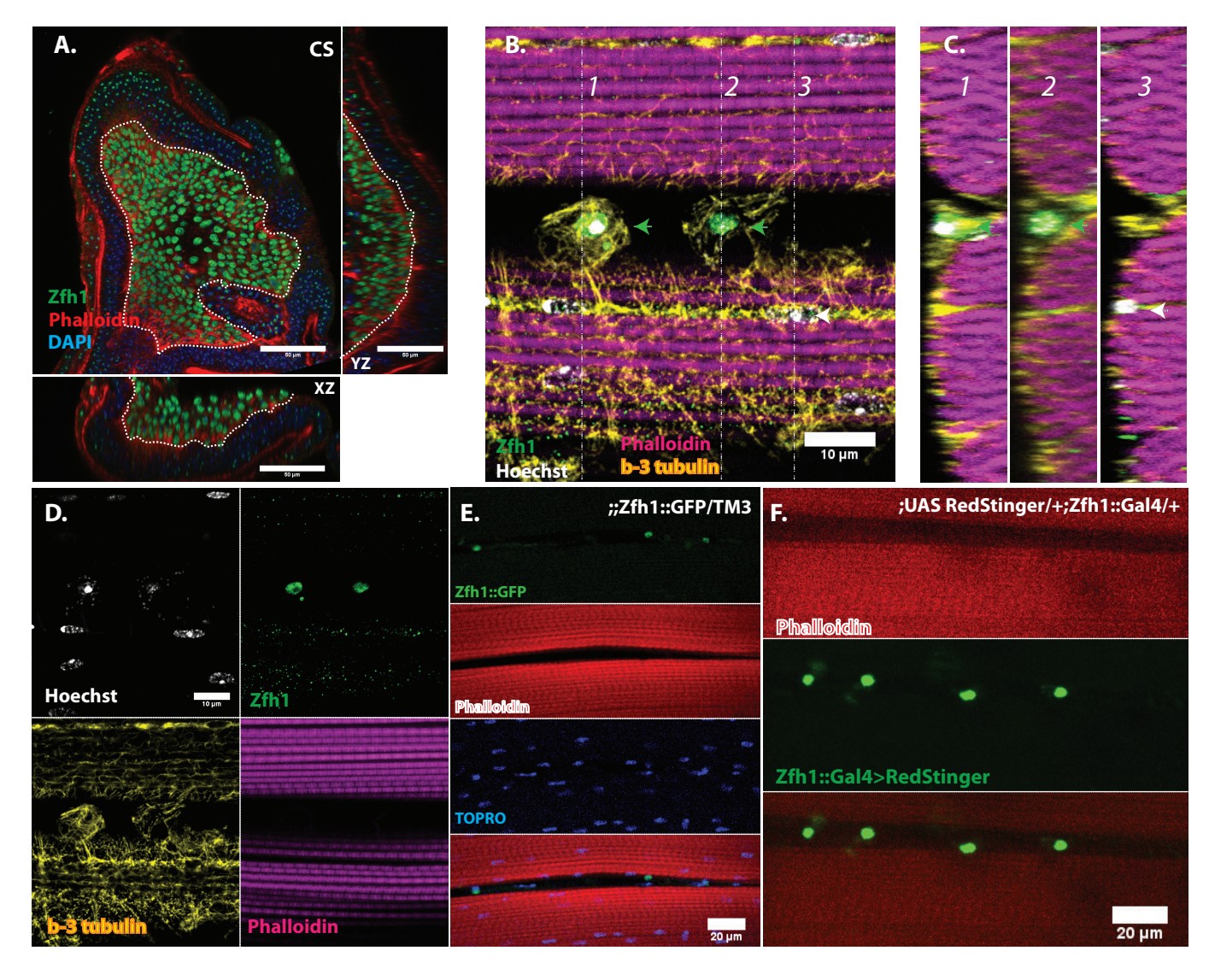

**Figure 4.** In adult muscle Zfh1 is a specific marker for unfused satellite cells. (A) Zfh1 immunolabeling (green) of third instar wing disc notum of wild-type flies. Zfh1 expression can be seen in myoblast nuclei located on the disc epithelium revealed by Phalloidin staining (red), as seen in XZ and YZ orthogonal views. TOPRO stains all nuclei (blue). Scale bar 50 um. N = 10. (B) Zfh1 and β−3-tubulin co-immunolabeling of adult DLMs in wild-type flies. Zfh1 expressing nuclei (green) co-stained with Hoechst (white) are located between DLM fibers labeled with Phalloidin (magenta). Two Zfh1 expressing nuclei are marked with green arrowheads. Nuclei inside DLMs do not express Zfh1 (one example indicated with white arrowhead). The cell boundaries of the DLMs and the Zfh1 expressing cells are delimited by β−3-tubulin; the Zfh1 expressing cell with its cytoskeleton is clearly separate from the adjacent DLM fibers. N = 25. (C) Three orthogonal views of the same preparation as in (B) taken at planes 1, 2 and 3 (dotted lines in B) document the positions of Zfh1 expressing cells outside the muscle fiber (green arrows). Their position contrasts with that of the fused Zfh1-negative DLM nuclei near the surface, one of which is indicated (white arrow). N = 15. (D) Same preparation as in (B) with montage showing the individual confocal channels for Hoechst staining (top left), Zfh1 immunolabeling (top right), β−3-tubulin immunolabeling (bottom left) and Phalloidin staining (bottom right). Scale bar 10 um. (E) Expression of GFP-tagged Zfh1 protein (green) in adult muscle of Zfh1::GFP/TM3 flies co-stained with Phalloidin (red) and TOPRO (blue). Top three panels show individual confocal channels; bottom panel is a superposition of the individual channels. Zfh1 protein expression is limited to unfused cells and is not seen inside muscle fibers. Scale bar 20 um. N = 15. (F) Expression of Zfh1-Gal4 (green) in adult muscle of UAS-RedStinger/+; Zfh1:Gal4/+flies co-labeled with Phalloidin (red). Top two panels show individual confocal channels; bottom panel is a superposition of the individual channels. Zfh1-Gal4 expression is limited to unfused cells and is not seen inside muscle fibers. N = 15. Scale bar 20 um.

DOI: https://doi.org/10.7554/eLife.30107.008

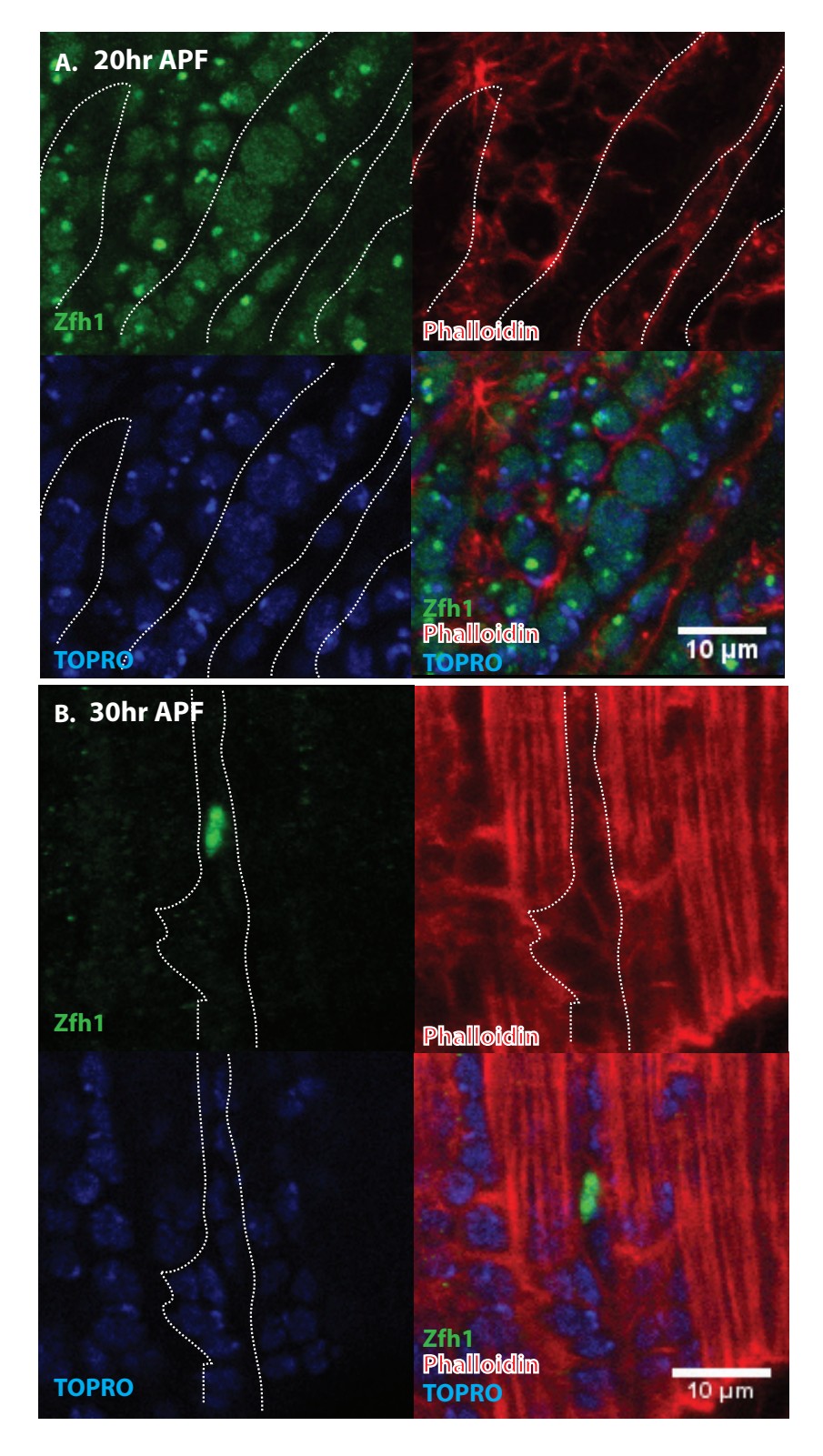

**Figure 5.** Pattern of Zfh1 expression in AMP lineal cells at 20 hr and 30 hr APF. Zfh1 immunolabeling (green) of unfused myoblasts and of myoblasts that have fused with DLM templates, co-labeled with Phalloidin (red) and TOPRO (blue). Zfh1 is expressed in all AMP lineal myoblasts at 20 hr APF but is restricted to a small set of unfused AMP lineal cells at 30 hr APF. (**A**) At 20 hr APF, Zfh1 expression is seen in all unfused myoblast nuclei and in all

*Figure 5 continued on next page*

*Figure 5 continued*
nuclei inside the DLM templates (outlined by white dotted line). N = 8. (**B**) At 30 hr APF, Zfh1 expression is limited to a few nuclei located in between DLM templates and is no longer seen in fused nuclei. Single example shown here. N = 10. Scale bars 10 um.
DOI: https://doi.org/10.7554/eLife.30107.009

## Muscle injury results in the proliferative expansion of the satellite cell population and the generation of fusion competent myoblasts

Vertebrate satellite cells are essential for muscle regeneration and repair in that muscle damage results in proliferative activity of satellite cells and the production of myoblasts that help rebuild compromised muscle tissue (*Mauro, 1961*; *Brack and Rando, 2012*; *Relaix and Zammit, 2012*). To investigate if satellite cells in *Drosophila* can also respond to muscle injury by increased proliferate activity, we induced physical damage in adult flight muscles mechanically and subsequently probed the damaged muscle for proliferative activity in the satellite cell population.

To induce muscle damage, localized stab injury of DLMs was carried out in adult flies using a small needle; care was taken to restrict damage such that only 1 or 2 muscle fibers were affected (*Figure 7*). DLMs damaged in this way can regenerate. While damage is still clearly evident 2 days after injury, significant morphological regeneration is manifest after 5 days, and after 10 days regeneration has progressed such that only small remnants of the injury are apparent (*Figure 7*)

To determine if muscle damage results in proliferative expansion of satellite cells, we compared the number of Zfh1-Gal4 labeled cell nuclei in uninjured control muscles versus injured muscles 24 hr after damage using UAS-nlsRedStinger. A dramatic increase in the number of Zfh1-labeled nuclei was seen in the damaged muscle as compared to controls (*Figure 8A, B*). This increase in Zfh1-positive nuclei number was strongest in the damaged muscle fibers and less pronounced in neighboring undamaged fibers. In contrast, in the damaged muscle fiber, increases in the number of Zfh1-labeled nuclei were seen along the entire extent of the fiber length. As expected, the Zfh1 labeled nuclei in the damaged muscle, as in controls, were largely located at the surface of muscle fibers; few, if any, of the Zfh1 labeled nuclei observed in these experiments were located within the injured muscle fiber. Interestingly, and in contrast to the situation in uninjured controls, many of the numerous Zfh1-labeled nuclei associated with the injured muscle appear to be manifest as spatially adjacent couplets (*Figure 8C, D*).

Taken together, these findings indicate that physical damage leads to a marked proliferative expansion in the satellite cell population associated with the injured flight muscle fiber. Given that the satellite cell population undergoes proliferative expansion following injury of adult muscle fibers, might some of their lineage correspond to cells that can fuse with the damaged muscle?

To investigate this, we used the Zfh1-Gal4, Gal80ts driver in G-trace experiments. G-trace (Gal4 technique for real-time and clonal expression) is a dual color genetic labeling technique based on Gal4 activity (*Evans et al., 2009*). The reporters used are RFP (red fluorescent protein) and GFP (green fluorescent protein), where RFP expression is strictly dependent on ongoing real-time Gal4 activity, while GFP expression is lineally dependent on previous Gal4 activity but independent of ongoing Gal4 activity.

G-trace labeling was induced in 1- to 3-day-old adults for 72 hr before muscle injury and recovered at various times (24 hr, 48 hr, 1 week) after muscle injury (*Figure 9*). At 24 hr after muscle injury, most of the labeled cell nuclei were both RFP positive and GFP positive (i.e. yellow), signifying both real-time and lineage-dependent previous activity of the Zfh1-Gal4 driver in these cells. Moreover, as expected for Zfh1-labeled cells in adult muscle, all of these were located at the muscle fiber surface (see above). Similar findings were obtained at 48 hr after muscle injury; both RFP-positive and GFP-positive cell nuclei were manifest at the muscle surface, while none were seen inside the bulk of DLMs. In contrast, 1 week after injury, labeled cells that appeared to be located inside muscle fibers were observed. Thus, in addition to RFP-positive and GFP-positive cell nuclei located at the surface of the muscle, GFP-positive nuclei indicative of lineage-specific previous activity of the Zfh1-Gal4 driver appeared to be positioned inside muscle fibers, albeit very close to their surface. Interestingly, the shape of these nuclei appeared to be flattened or disc-like in contrast to the round shape of the nuclei located outside of the muscle cell surface.

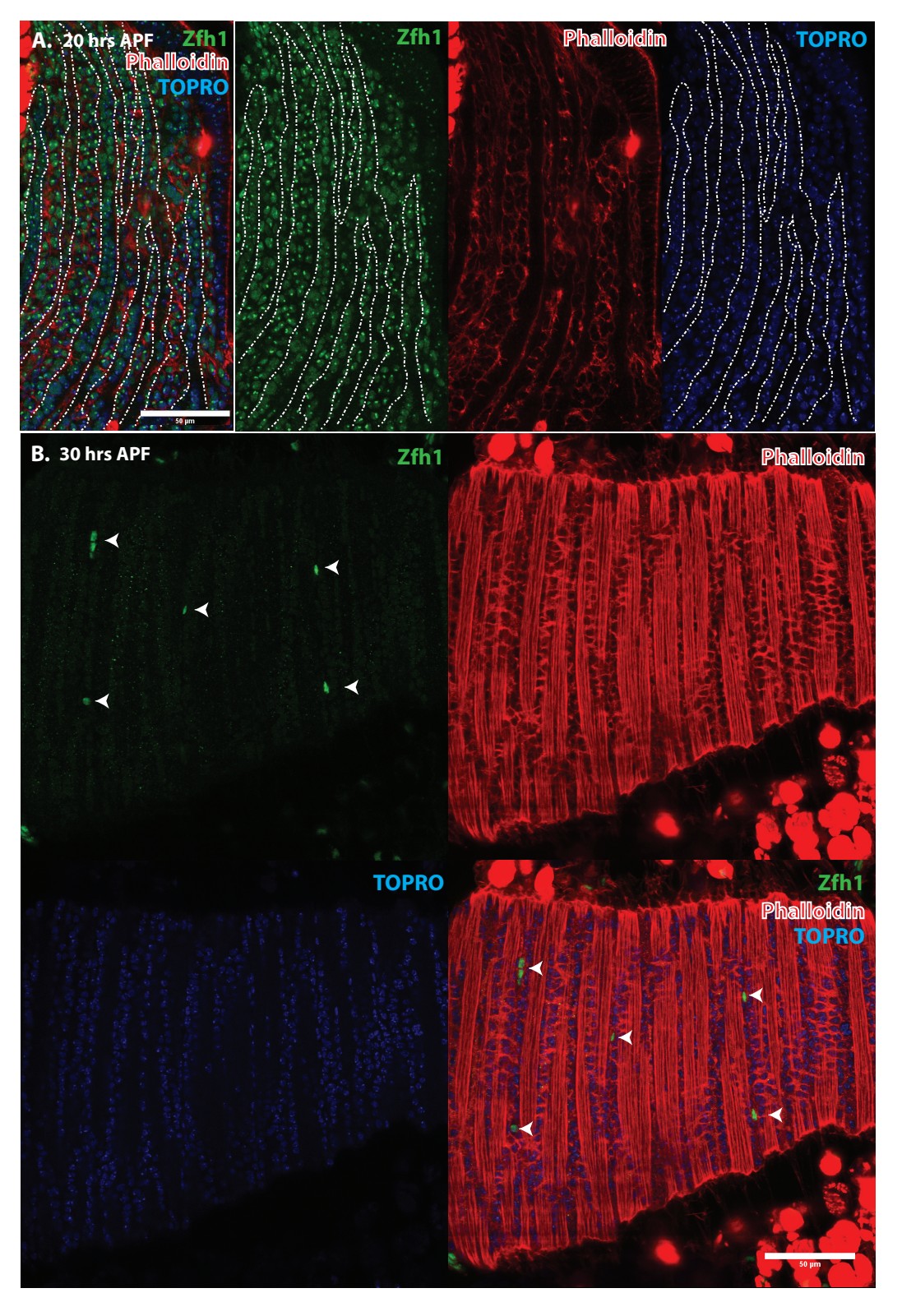

**Figure 6.** Pattern of Zfh1 expression in AMP lineal cells at 20 hr and 30 hr APF. Zfh1 immunolabeling (green) of unfused myoblasts and of myoblasts that have fused with DLM templates, co-labeled with Phalloidin (red) and TOPRO (blue). Zfh1 is expressed in all AMP lineage myoblasts at 20 hr APF but is restricted to a small set of unfused AMP lineage cells at 30 hr APF. (A, B) As in *Figure 5* but at lower magnification with images showing whole templates at respective time points. Scale bars 50 µm.

*Figure 6 continued on next page*

*Figure 6 continued*

DOI: https://doi.org/10.7554/eLife.30107.010

In uninjured adult muscle, nuclei of the Zfh1 lineage are always located at the muscle cell surface and are never seen inside the muscle fiber (see above). Hence, the possibility that nuclei of the Zfh1 lineage might be located inside the muscle 1 week after injury implies that the corresponding Zfh1 lineal cells have fused with the muscle fiber. To investigate the possibility that Zfh1 lineal progeny might have fused with mature DLMs after injury, we repeated these G-trace experiments in the background of alpha Spectrin immunolabeling. Alpha Spectrin is a plasma-membrane-associated protein that marks muscle cell boundaries and can also be seen at low levels in the cytoplasm of DLMs (*LaBeau-DiMenna et al., 2012*).

In age matched, uninjured animals with G-trace induced in the adult stage, Zfh1 lineage cell nuclei that are both RFP positive and GFP positive can be seen closely associated with surface of the DLM as delimited by alpha Spectrin labeling but clearly located outside of the muscle cell (*Figure 10A*). In contrast, in animals 1 week after muscle injury, nuclei of Zfh1 lineage cells that are GFP labeled are located within the muscle fiber albeit near its surface (*Figure 10B*). Remarkably, these nuclei have a flattened, disc-like shape and appear to be markedly larger than those of uninjured controls. Similar findings are obtained in comparable G-trace experiments in which Vinculin immunolabeling is used to demarcate the muscle fiber surface. In uninjured control muscle fibers, RFP- and GFP-positive Zfh1 lineage nuclei are located at the outer surface of the muscle fiber (*Figure 10C*). In muscle fibers 1 week after injury, GFP labeled Zfh1 lineage cell nuclei are positioned inside the muscle cell but remain near the muscle cell surface (*Figure 10D*).

Taken together, these findings imply that following muscle injury, cells of the Zfh1 lineage, that is, lineal descendants of the Zfh1-expressing satellite cells, fuse with the damaged muscle fibers. This fusion process, which is preceded by a proliferative expansion of the normally quiescent satellite cell population, may contribute to the promotion of muscle fiber repair and, hence, represent a remarkable functional similarity in the role of satellite cells in response to muscle injury in flies and vertebrates.

## Proliferative activity of satellite cells in response to muscle injury requires Notch expression in satellite cells and Delta expression in muscle fibers

It has previously been shown that proliferative activity of AMPs during development requires Notch signaling (*Gunage et al., 2014*). Might the AMP lineal descendant satellite cells in adult muscle also require Notch signaling for injury-induced proliferative activity? To investigate this, we first determined if satellite cells express Notch. For this, we used a Notch-Gal4 driver in G-trace experiments. In uninjured controls, all the nuclei within the muscle fiber were GFP-positive, in accordance with their lineal origin from Notch expressing AMPs, but none were RFP positive (*Figure 11A*). In contrast the muscle surface associated nuclei were RFP positive due to real-time activity of the Notch-Gal4 driver in implying that the nuclei of satellite cells express Notch. To establish that the RFP-positive nuclei were indeed the nuclei of satellite cells, we combined these G-trace experiments with Zfh1-immunolabeling. In these experiments the RFP-positive nuclei at the muscle fiber surface were always Zfh1-positive, while the GFP-positive nuclei within the muscle fiber were Zfh1-negative (*Figure 11B*). These findings indicate that satellite cells in intact muscle fibers express Notch.

Following muscle injury, an expansion of the satellite cell lineage takes place (see above). To determine if the cells in this expanded lineage continue to express Notch, we repeated the G-trace experiments 24 hr after muscle injury. In these experiments, as in the uninjured control, all of the muscle surface associated nuclei were RFP positive due to real-time activity of the Notch-Gal4 driver implying that the nuclei of the expanded satellite cell population continue to express Notch (*Figure 11C*). Moreover, quantification of the number of Notch-Gal4 expressing cell nuclei in injured muscle fibers versus uninjured controls indicates that an approximately twofold expansion in the number of Notch expressing satellite cells occurs 24 hr after muscle injury (*Figure 11D*).

We next investigated if muscle fibers might express the Notch ligand Delta. Immunolabeling of uninjured flight muscles revealed a significant albeit low level of Delta expression (*Figure 12A*). In

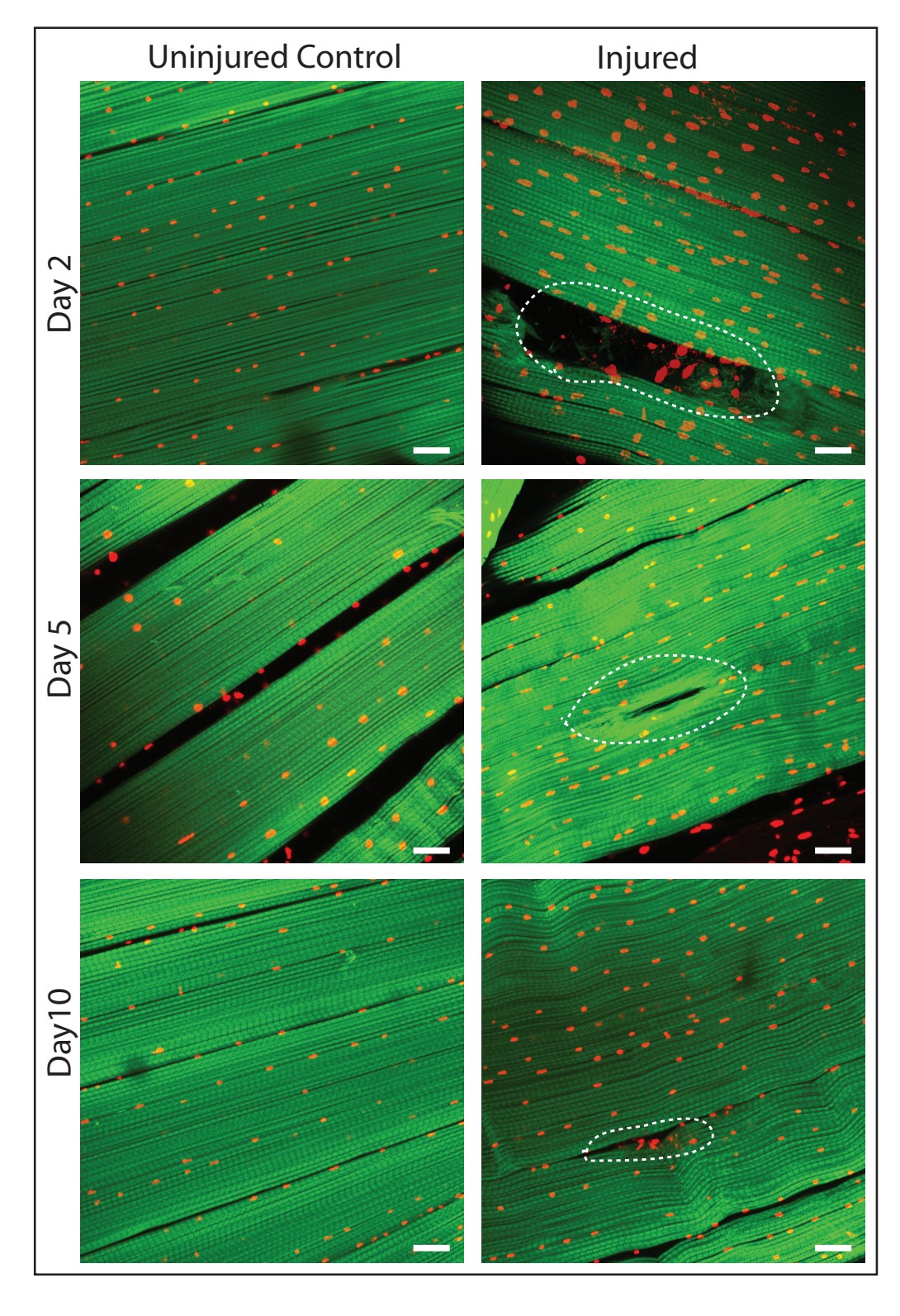

**Figure 7.** DLM fibers regenerate following induced physical damage. Representative images of injured flight muscles (right) and time matched controls (left) at day 2, 5 and 10 after localized stab injury. Adult DLMs stained with Phalloidin (green) and TOPRO (red). At day 2 following injury, breaks in actin filaments, and corresponding disruptions in distribution of nuclei at the site of the injury wound (indicated by white dotted line) are seen. At day 5

*Figure 7 continued on next page*

*Figure 7 continued*
following injury, the wound is reduced in size and the actin filament arrangement and myonuclei distribution is more has recovered. At day 10 following injury, regeneration is virtually complete and only small remnants of the wound are apparent. N = 10/group per time point. Scale bar 15 um.
DOI: https://doi.org/10.7554/eLife.30107.011

contrast, a dramatic increase in Delta expression was observed in injured flight muscles (*Figure 12B*). Indeed, a quantification of the intensity of immunolabeling in control versus injured muscles indicates that a fourfold increase in Delta expression occurs in response to injury (*Figure 12C*). A comparable upregulation of Neuralized, an E3-ubiquitin ligase required in the Delta-Notch signal transduction process for Delta endocytosis, was also observed in damaged muscle versus controls (*Figure 12D, E*). Analysis of a Neuralized-LacZ reporter line indicates that the muscle fiber-specific expression of Neuralized is significantly up-regulated following muscle injury (*Figure 12F*).

Taken together these findings indicate that following muscle injury, Notch is expressed throughout the expanding satellite cell population. Moreover they indicate that Delta expression is upregulated in the injured muscle fibers. Might signaling between muscle fiber associated Delta ligand and satellite-cell-associated Notch receptor be required for the proliferative mitotic activity of satellite cells in response to muscle injury?

To investigate this possibility, we used the mitotic marker phosphohistone-H3 (PH-3) on injured flight muscle. PH-3 labeling was carried out 12 hr after muscle injury. In injured wild-type controls, numerous satellite cells were PH-3 positive, indicative of the extensive mitotic activity associated with injury-induced satellite cell expansion (*Figure 13A*).

In contrast, in Act88F-driven (muscle-specific) Delta-RNAi knockdown experiments, limited to adult stages by Gal80-ts, a dramatic reduction in the number of PH-3 labeled satellite cells was observed in injured muscles as compared to controls (*Figure 13B*). A quantification of this reduction in PH-3-labeled satellite is shown in *Figure 13C*. Comparable results were obtained when a dominant negative form of Delta was expressed using the Act88F driver in injured muscle fibers (*Figure 13C*). These findings indicate that Delta expression in muscle fibers is required for the induction of injury-dependent mitotic activity of satellite cells.

To determine if Notch expression in satellite cells is similarly required for injury induced mitotic activity of satellite cells; comparable PH-3 labeling experiments were carried out using a temperature sensitive Notch allele. A quantification of the number of PH-3-labeled cells in the injured muscle of Notch temperature-sensitive allele flies at permissive (17°C) versus restrictive (29°C) temperature is shown in *Figure 11D*. While numerous satellite cells were PH-3 positive at the permissive temperature, at the restrictive temperature only few cells were PH-3-positive. (Similar results were obtained by using the chemical inhibitor DAPT, a gamma-Secretase inhibitor, to block Notch pathway activity; data not shown). This finding was confirmed in Dmef2-driven Notch-RNAi knockdown experiments, in which the knockdown was restricted to adult stages via Gal80-ts repressor; a dramatic reduction of PH-3-labeled cell number in injured muscle fibers as compared to controls was observed (*Figure 13E*). These findings imply that functional Notch expression in the satellite cells is indeed required for injury-induced proliferation of the satellite cells population

Taken together with the previously mentioned experiments, our findings are in accordance with a model in which lineal descendants of muscle stem cell-like AMPs are present in adult muscle as muscle fiber apposed satellite cells. Although normally quiescent, following muscle fiber injury these satellite cells become mitotically active, engage in Notch-Delta signaling-dependent proliferative activity and generate lineal descendant cell populations, which can fuse with the injured muscle fiber.

## Discussion

The identification and characterization of satellite cells in *Drosophila* indicates that muscle stem cell lineages act not only in the development of flight muscle as reported previously (*Gunage et al., 2014*), but also have a role in the mature muscle of the adult. Thus, comparable to the situation in vertebrates, the Drosophila satellite cells are lineal descendants of the muscle-specific stem cell-like AMPs generated during embryogenesis, become intimately associated with adult muscle fibers and

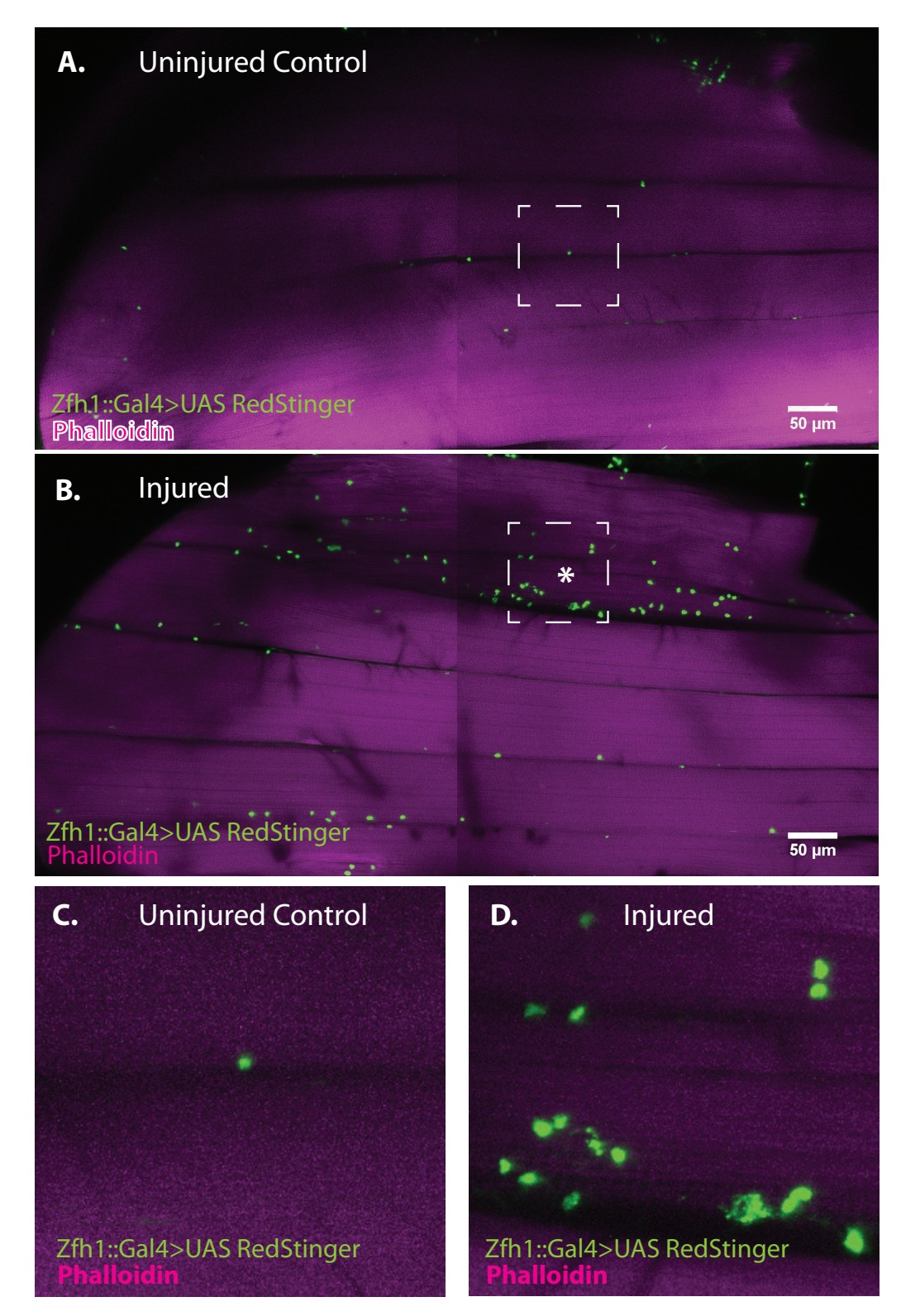

**Figure 8.** Zfh1-positive satellite cells located between DLMs proliferate in response to physical injury. (A) Uninjured control DLMs. Zfh1-Gal4 driving UAS RedStinger in adult DLMs. A small number of Zfh1-Gal4-positive mononucleate satellite cells (green) are located between DLM fibers. Anti-DsRed (green) co-stained with Phalloidin (magenta). Scale bar 50 μm. (B) Injured DLMs. Zfh1-Gal4 driving UAS RedStinger in adult DLMs injured by stab wound (* denotes injured fiber). At 24 hr after stab wound, numerous Zfh1-Gal4-positive mononucleate satellite cells (green) are seen between DLM fibers at

*Figure 8 continued on next page*

*Figure 8 continued*

the site of injury but also away from the site of injury. Anti-DsRed (green) co-stained with Phalloidin (magenta). Scale bar 50 µm. (C) Single Zfh1-Gal4-labeled nucleus located between uninjured DLM fibers in area delineated by white square in A. (D) Multiple doublets and a few singlets of Zfh1-Gal4-positive nuclei located near site of injury in area delineated by white square in (B). N = 15 Scale bar 50 µm.

DOI: https://doi.org/10.7554/eLife.30107.012

remain quiescent under normal circumstances. Following muscle injury, these Zfh1-expressing cells engage in Notch-Delta signaling-dependent proliferative activity and generate lineal descendant progeny that can fuse with the injured fibers. With improved immune-EM staining protocols on *Drosophila* DLMs, visualizing Zfh1 expression in these cells at the ultrastructural level will prove valuable.

Previous work has shown a role of Zfh1 in other developmental and maintenance processes in *Drosophila*. Thus, loss of Zfh1 function leads to defects in *Drosophila* somatic muscle, heart and gonad development (*Sellin et al., 2009*; *Lai et al., 1993*). Adult germline stem cells in *Drosophila* testes, require Zfh1 for stem cell maintenance (*Leatherman and Dinardo, 2008*). Zfh1 is also known to control neural lineages in a Notch-dependent manner (*Lee and Lundell, 2007*; *Garces and Thor, 2006*; *Su et al., 1999*). In addition, Zfh1 acts near the top of a transcription factor cascade that influences hematopoesis beginning in embryos and continuing in larvae (*Frandsen et al., 2008*). Evidently, Zfh1 expression is associated with the development, maintenace and differentiation of stem cells in multiple tissues across the lifespan of *Drosophila*.

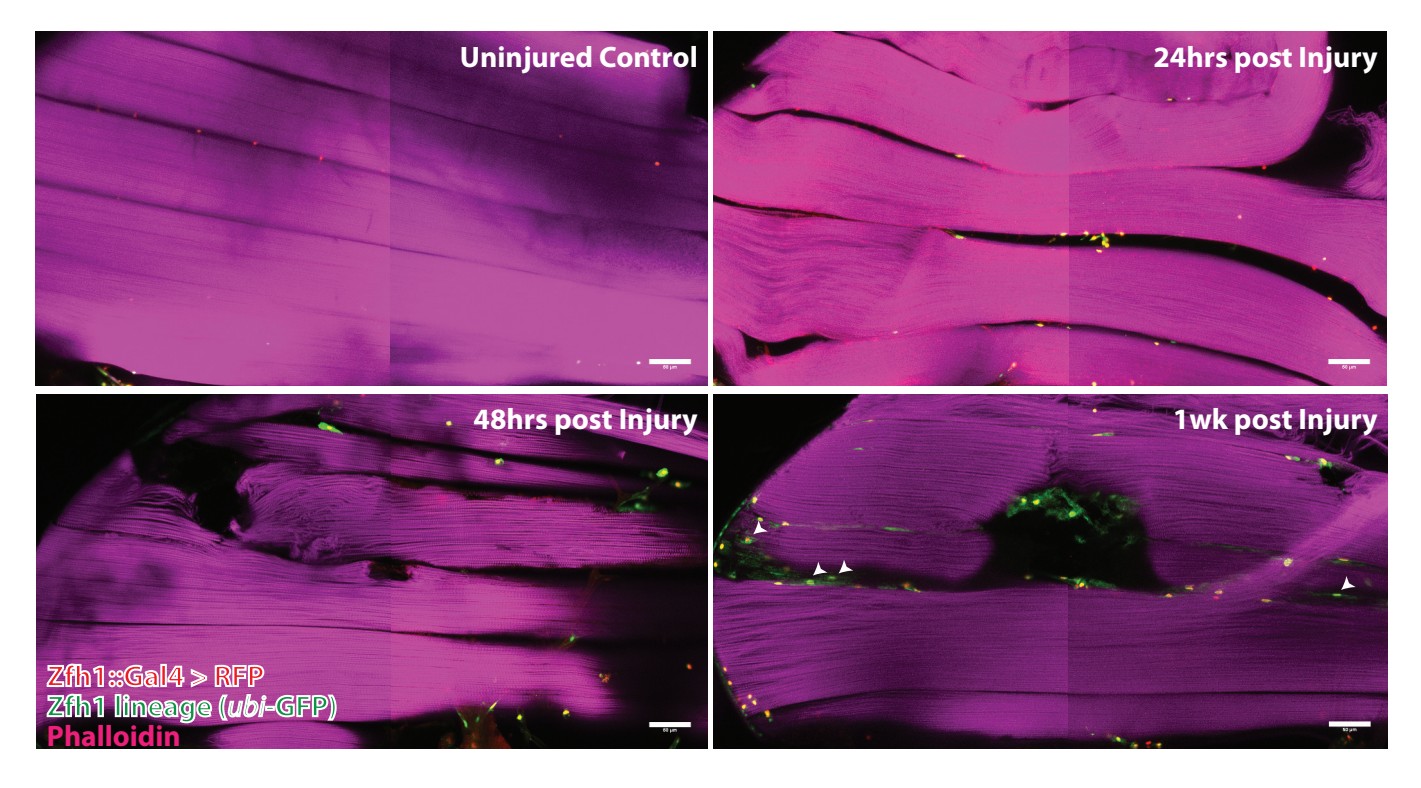

**Figure 9.** Following muscle injury, satellite cell lineal progeny localize to the surface and the interior of DLM fibers. Localization of satellite cell lineal progeny examined with Zfh1-Gal4 driven G-trace labeling in uninjured control and in DLM muscles at 24 hr, 48 hr and 1 week after injury. In all cases, G-trace was induced in the adult stage and in the injured animals, 24 hr before infliction of a stab wound. In the uninjured control, only a few cell nuclei located at the DLM surface are labeled as expected for Zfh1-positive satellite cells (top left). At 24 hr and 48 hr after injury more cell nuclei located at the DLM surface are labeled indicating proliferative expansion of the Zfh1-positive satellite cell lineage (top right, bottom left). At 1 week after injury, labeled cell nuclei are located both at the surface and in the interior of DLM fibers implying that some of the Zfh1-positive satellite cell lineal descendants have now fused with the injured DLMs (bottom right). N = 6 per group Scale bar 50 um.

DOI: https://doi.org/10.7554/eLife.30107.013

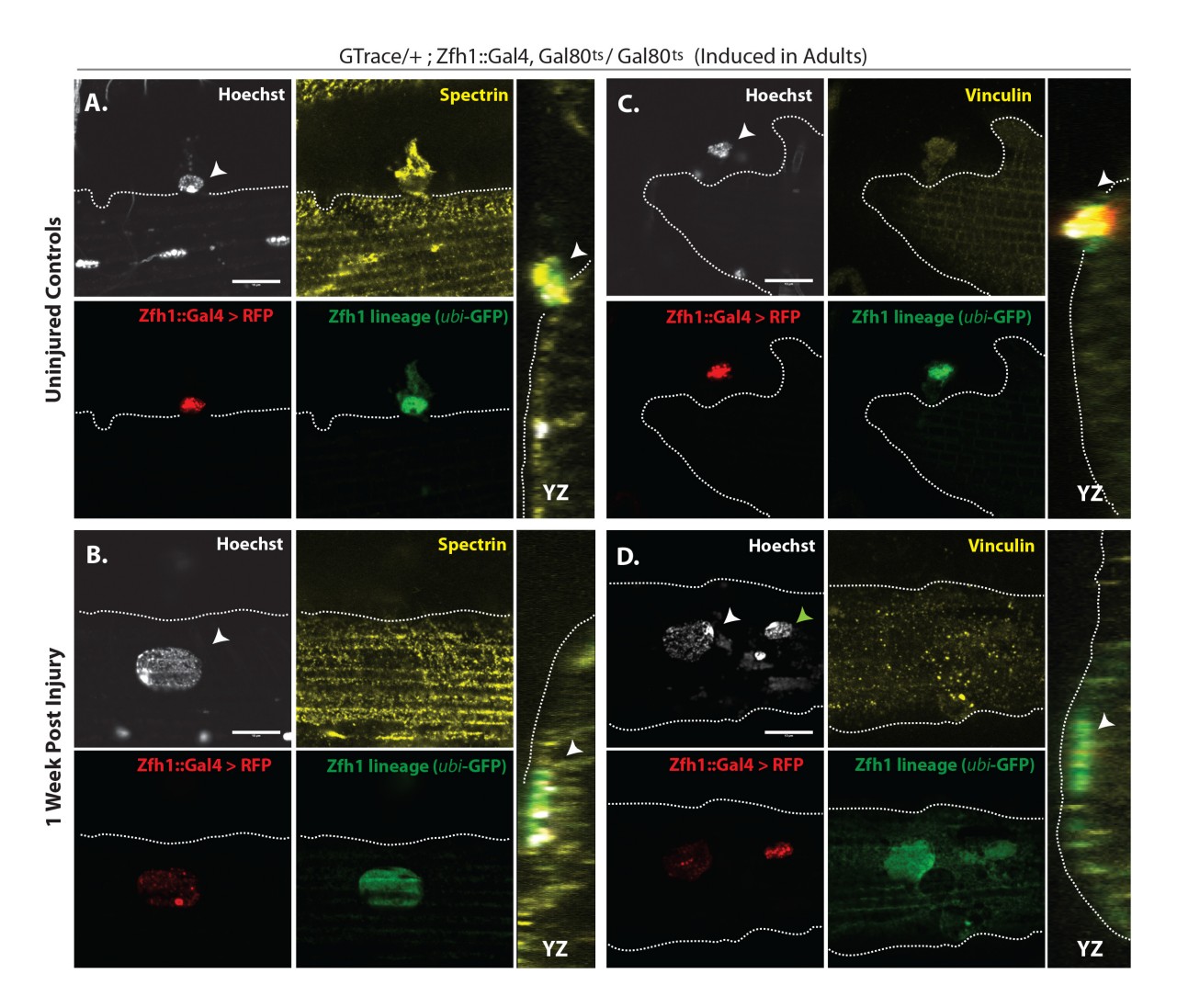

**Figure 10.** G-trace labeled lineal progeny of Zfh1 expressing satellite cells fuse with damaged DLM fibers after injury. Localization of satellite cell lineal progeny examined with Zfh1-Gal4 driven G-trace in uninjured controls (**A, C**) and in experimental animals 1 week after injury (**B, D**). G-trace was induced in the adult stage and in the injured animals. This induction was 72 hr before infliction of a stab wound. In (**A–D**), the left four panels are individual confocal channels for Hoechst staining (top left), alpha Spectrin or Vinculin staining (top right), G-trace driven RFP expression (bottom left) and G-trace driven GFP expression (bottom right); the right panel is a superposition of the four channels and viewed from an orthogonal YZ perspective. (**A**) Control. A single-cell nucleus closely associated with the outside of the DLM surface, as delineated by the alpha Spectrin expression border (dotted line), is both RFP labeled (implying real-time Zfh1 expression) and GFP labeled (implying lineal origin from a Zfh1-positive cell) indicating that it corresponds to a Zfh1-expressing satellite cell. (**B**) Injured. A single-cell nucleus located within the muscle fiber albeit close to the fiber's surface, as delineated by the alpha Spectrin expression border (dotted line) is GFP labeled (implying lineal origin from a Zfh1-positive cell) indicating that it corresponds to a satellite cell lineal progeny. The labeled nucleus has a flattened disc-like shape. (**C**) Control. A single-cell nucleus closely associated with the outside of the DLM surface as is delineated by the Vinculin expression border (dotted line), is both RFP labeled (implying real-time Zfh1 expression) and GFP labeled (implying lineal origin from Zfh1-positive cells) indicating that it corresponds to a Zfh1-expressing satellite cell. (**D**) Injured. A single-cell nucleus located within the muscle fiber albeit close to the fiber's surface (white arrow), as delineated by the Vinculin expression border (dotted line) is GFP labeled (implying lineal origin from a Zfh1-positive cell) indicating that it corresponds to a satellite cell lineal progeny. The labeled nucleus has a flattened disc-like shape. Note that the second, apparently adjacent cell nucleus (green arrow) which is RFP labeled is not located within the muscle fiber. N = 8 per group. Scale bars 10 μm.

DOI: https://doi.org/10.7554/eLife.30107.014

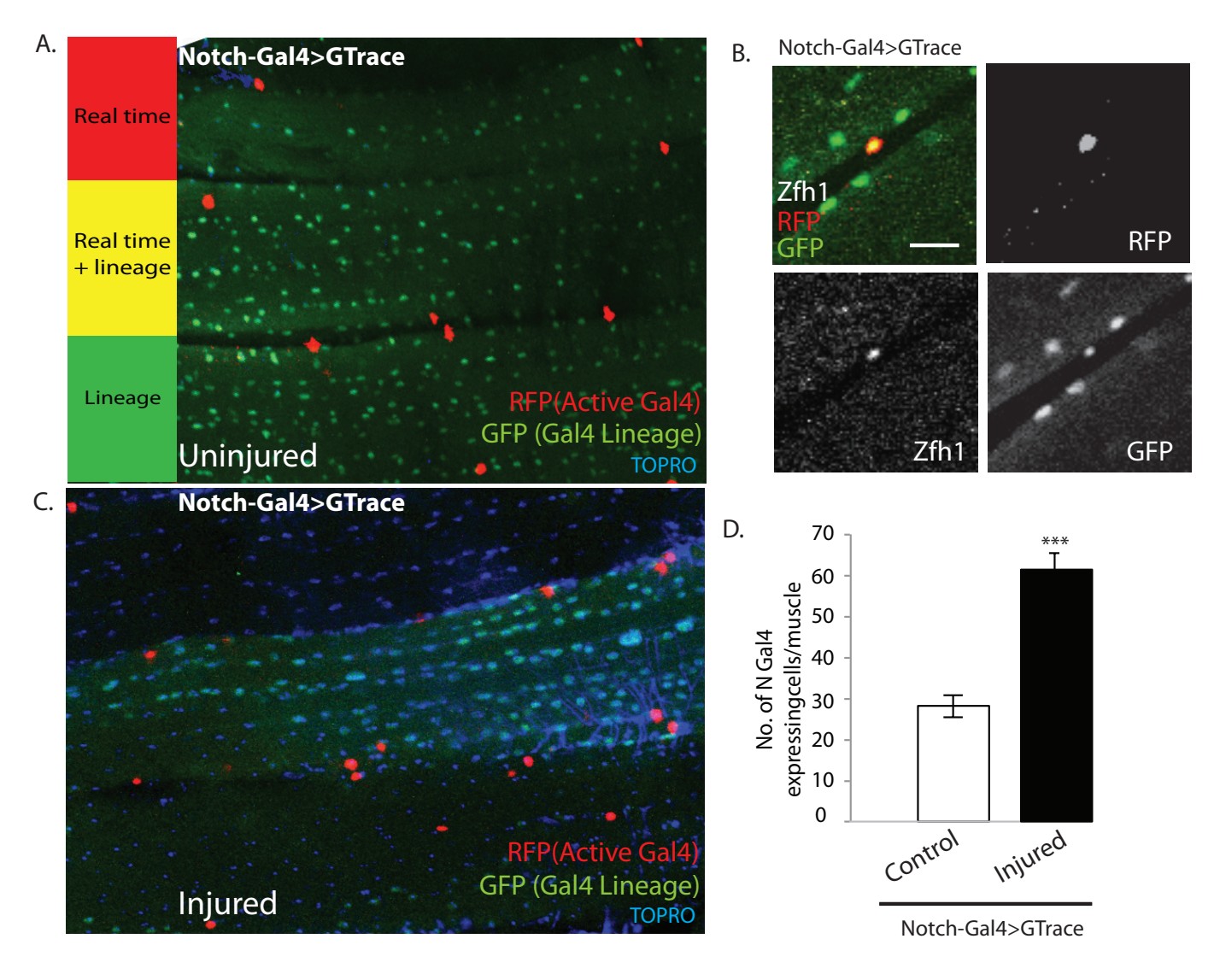

**Figure 11.** Notch-Gal4 driven G-trace labeling reveals real time Notch expression in muscle satellite cells. (**A**) Notch-Gal4 driven G-trace labeling of uninjured adult DLM co-stained by TOPRO. The cell nuclei located on the surface of the DLM fibers are RFP positive (red, anti-RFP labeled) indicating that they correspond to satellite cells that are actively expressing Notch. In contrast, the numerous myonuclei within the muscle fiber are GFP positive (green, anti-GFP labeled) confirming the fact that they are lineal descendants of Notch expressing AMPs. (**B**) Notch-Gal4 driven G-trace labeling of uninjured adult DLM co-labeled by Zfh1 immunostaining. The cell nuclei located on the surface of the DLM fibers show strong expression of RFP indicating real-time Notch-Gal4 expression and of Zfh1 indicating that they are satellite cells. The top left panel is a superposition of individual channels for Zfh1, RFP, and GFP expression; the remaining panels are the corresponding single channels. (**C**) Notch-Gal4 driven G-trace labeling of adult DLM 24 hr after injury, co-stained by TOPRO. The cell nuclei located on the surface of the injured DLM fibers have increased in number but are still RFP positive (red, anti-RFP labeled) indicating real-time expression of Notch-Gal4 in these nuclei of the expanded satellite cell lineage. As in (**A**), myonuclei within the muscle fiber are GFP positive (green, anti-GFP labeled). (**D**) Quantification of the number of real-time Notch-Gal4 expressing nuclei in control versus injured muscle fibers in these G-trace experiments. Twice as many RFP-positive cells are observed in injured versus control DLMs. n = 10 Data presented are mean ± standard error Student's t test: p-value<0.001 ***. Scale bar 10 μm.
DOI: https://doi.org/10.7554/eLife.30107.015

The mammalian homologs of Zfh1 known as ZEB1 and ZEB2 have been largely studied for their role in cancer progression (*Vandewalle et al., 2009*). Moreover, the ZEB family protein ZEB2, controls adult hematopoetic differentiation in adults (*Li et al., 2017*; *Postigo and Dean, 2000*). Direct functional analyses of ZEB, in mammalian muscles are restricted to C2C12 cells (*Postigo and Dean, 1997*; *Fontemaggi et al., 2001*). In all cases, ZEB expression was found to inhibit differentiation of

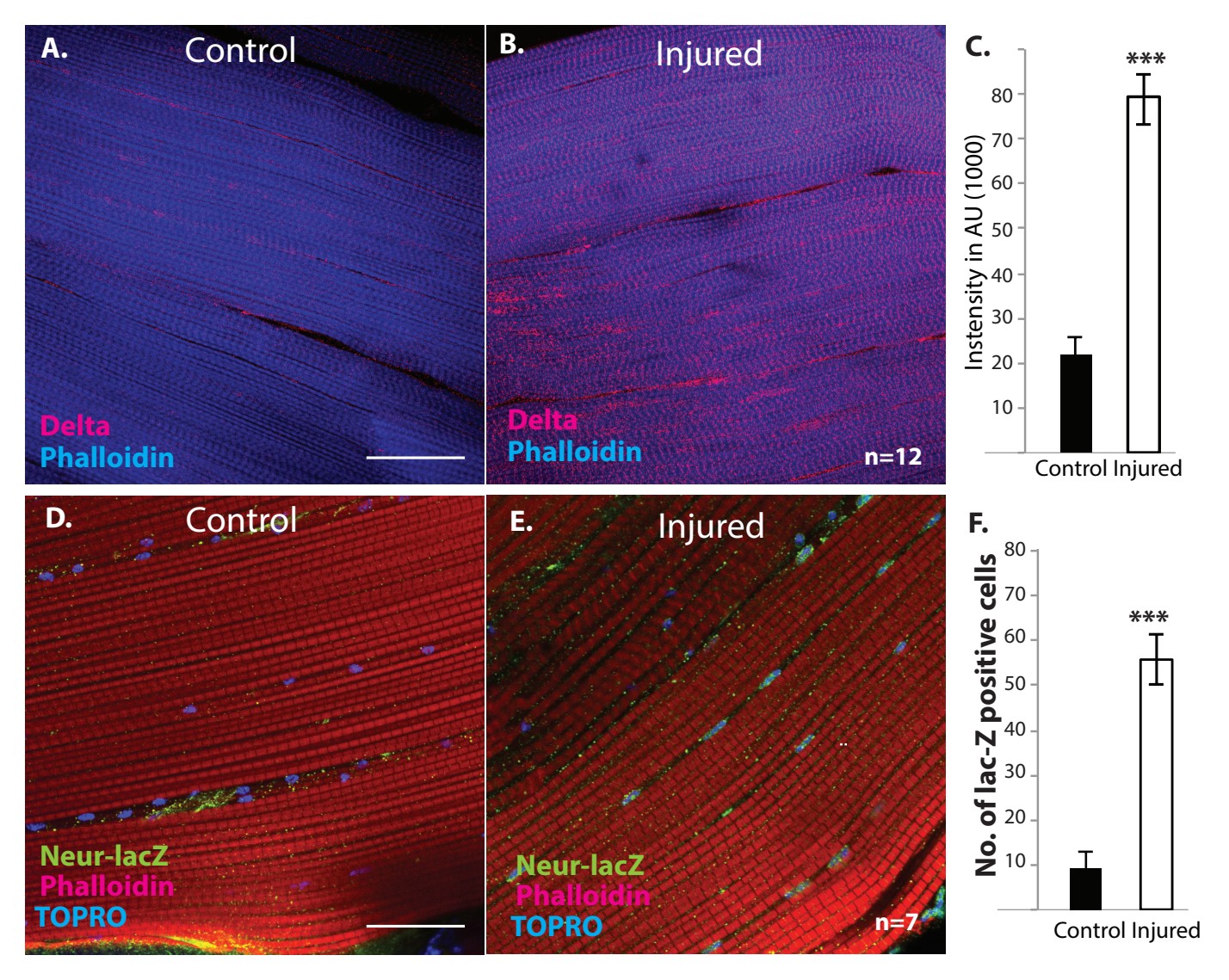

**Figure 12.** Delta and Neuralized are upregulated in injured muscle fibers. (A, B) Delta-GFP (anti-GFP, red) expression in DLMs co-labeled with Phalloidin (blue) in control (A) versus injured (B) muscle fibers reveals a marked upregulation of Delta-GFP expression upon injury. (C) Quantification of signal intensity of Delta-GFP in control versus injured DLM fibers; injured muscles show significant upregulation of Delta expression in comparison to uninjured muscle (quantification in arbitrary intensity units). n = 12 Data presented are mean ± standard error. Student's t test: p-value<0.001 ***. (D, E) Neuralized-LacZ expression (green, anti-LacZ immunolabeling) co-labeled by Phalloidin (red) in control versus injured muscle fibers. In comparison to controls (D), injured muscles show elevated Neuralized-lacZ levels in myonuclei, some of which are indicated by white arrows in (E). (F) Quantitation of Neuralized-LacZ expression in control versus injured muscle. For quantification the number of lac-Z-positive nuclei were counted. n = 7. Data presented are mean ± standard error. Student's t test: p-value<0.001 ***. Scale bars 30 μm.

DOI: https://doi.org/10.7554/eLife.30107.016

myoblasts. In view of these results, our findings in Drosophila motivate a deeper look into ZEB function in muscle development in mammals.

The remarkable similarities in lineage, structure and function of satellite cells in flies and vertebrates imply that the role of these adult-specific muscle stem cells is evolutionarily conserved and, hence, is likely to be manifest in other animals as well. Recently, satellite cells have been identified in a crustacean (Parhyale hawaiensis) during limb regeneration (*Konstantinides and Averof, 2014*). It will now be interesting to determine if comparable satellite cells are also present in adult

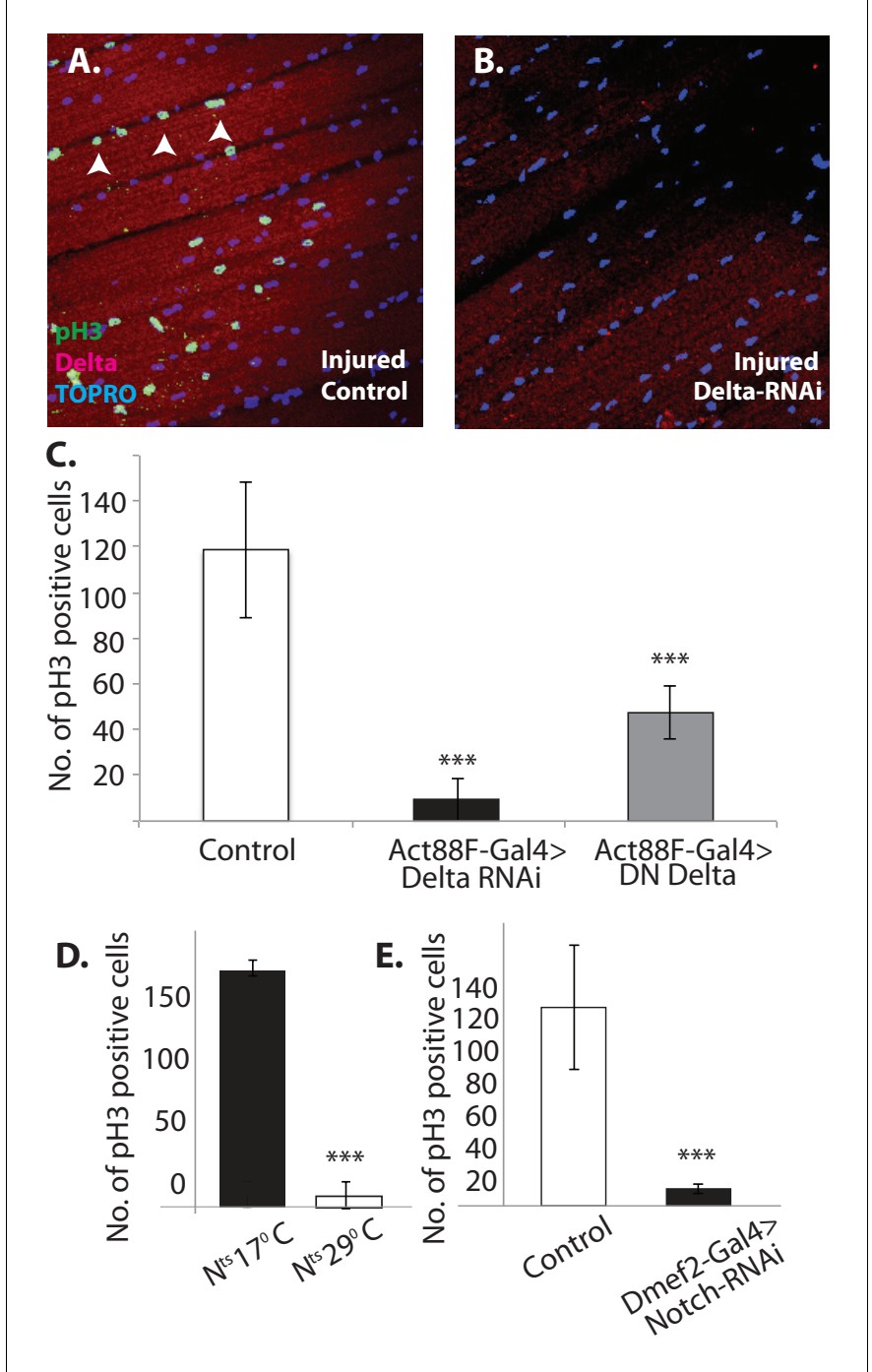

**Figure 13.** Notch-Delta signaling is required for satellite cell proliferative activity in injured muscle. (**A**) Injured control muscle. Mitotic activity assayed by PH-3 expression (green, anti-PH-3 immunolabeling) in DLMs co-labeled for Delta expression (red, anti-Delta immunolabeling) and TOPRO. Numerous satellite cells (three indicated by white arrows) are PH-3-positive indicative of the mitotic activity required for injury-induced expansion of the satellite cell population. (**B**) Injured muscle with adult-specific Delta downregulation (via Act88F-Gal4, TubGal80ts driving UAS Delta RNAi). Mitotic activity of satellite cells assayed by PH-3 expression as in (**A**) is absent. (**C**) Quantification of the number of PH-3 expressing cells in control versus Delta downregulated flies; Delta downregulation is achieved by targeted Delta-RNAi knockdown as well as by targeted dominant negative Delta (DN Delta) expression. n = 9 Data presented are mean ± standard error. Student's t test: p-value<0.001***. (**D**) Quantification of PH-3 labeled cells in injured muscle of Notch temperature sensitive allele flies at permissive (17°C) versus restrictive temperature (29°C). n = 12. Data presented are mean ± standard error. Student's t test:

*Figure 13 continued on next page*

*Figure 13 continued*

p-value<0.001***. (E) Quantification of mitotically active PH-3-labeled satellite cells in control versus Notch downregulated flies. n = 12. Data presented are mean ± standard error. Student's t test: p-value<0.001***.

DOI: https://doi.org/10.7554/eLife.30107.017

musculature of other key protostome and deuterostome invertebrate phyla such as molluscs, annelids and echinoderms.

In vertebrates, satellite cells can undergo symmetric divisions which expand the stem cell pool and asymmetric divisions in which they self-renew and also generate daughter cells that differentiate into the fusion-competent myoblasts required for muscle regeneration and repair (*Abmayr and Pavlath, 2012*; *Relaix and Zammit, 2012*). In *Drosophila*, symmetric and asymmetric division modes are seen during development in the muscle stem cell-like AMPs. Notch signaling controls the initial amplification of AMPs through symmetric divisions, the switch to asymmetric divisions is mediated by Wingless regulated Numb expression in the AMP lineage, and in both cases, the wing imaginal disc acting as a niche provides critical ligands for these signaling events (*Gunage et al., 2014*). It will important to determine if fly satellite cells, as lineal descendants of AMPs, manifest similar cellular and molecular features in their proliferative response to muscle injury and, thus, recapitulate myogenic developmental mechanisms in the regenerative response of adult muscle. It will also be important to investigate if the mature muscle acts as a niche in this process.

We show that proliferation of Zfh1 expressing cells in DLMs is rapid and extensive in response to stab wounds. We also show that first instances of fusion are seen removed in time from this burst of proliferation. Such a delay between satellite cell proliferation and fusion is puzzling and is also seen in vertebrates. We speculate that satellite cell fusion in this paradigm may serve as a later step in a regenerative process initiated soon after injury, possibly through signaling between injured muscle fibers and satellite cells. Further investigations characterizing the molecular and physical events in DLM repair are likely to clarify our understanding of this process and the system we have developed allows such directions to be explored.

*Drosophila* has proven to be a powerful genetic model system for unraveling the fundamental mechanisms of muscle development and stem cell biology, and in both respects many of the findings obtained in the fly have been important for the analysis of corresponding mechanisms in vertebrates (*Abmayr and Pavlath, 2012*; *Roy and VijayRaghavan, 1998*; *Egger et al., 2008*; *Homem and Knoblich, 2012*; *Jiang and Reichert, 2013*). With the identification of satellite cells in *Drosophila*, the wealth of classical and molecular genetic tools available in this model system can now be applied to the mechanistic analysis adult-specific stem cell action in myogenic homeostasis and repair. Given the understanding of various fusion molecules involved in early stages of myogenesis, it will also be interesting to investigate a possible conservation of the fusion molecular machinery for regeneration and repair in the adult (*Dhanyasi et al., 2015*; *Haralalka and Abmayr, 2010*). Finally, in view of the evidence for age and disease-related decline in satellite cell number and function in humans (e.g.*Chang and Rudnicki, 2014*), this type of analysis in *Drosophila* may provide useful information for insight into human muscle pathology.

## Materials and methods

### Fly strains, genetics and MARCM

Fly stocks were obtained from the Bloomington Drosophila Stock Centre (Indiana, USA) and were grown on standard cornmeal medium at 25°C.

For MARCM experiments mentioned in flies of genotype Hsflp/Hsflp; FRT 42B, Tub Gal80 were crossed to +; FRT 42B, UAS GFP/CyO Act-GFP; Dmef2-Gal4. For MARCM experiments, two heat shocks of 1 hr each separated by 1 hr were given to either late third instar larvae or young adults for clonal induction. Clones were either recovered in the late larval stage for wing disc analysis or in adult stages, which were dissected and processed for flight muscles.

In knockdown and overexpression experiments the following lines were used: +; +; Dmef2-Gal4, Gal80ts, Act 88 F-Gal4, Gal80ts, UAS Notch RNAi (Bloom, 35213), UAS Neur RNAi (Bloom, 26023), UAS DN Delta (Bloom, 26697), UAS Delta RNAi (VDRC, 37288 and GD3720).

Other stocks used- Dl-GFP (Bloom, 59819), Neur-LacZ (Bloom, 12124), Bloom-55121 and 55122. G-trace analysis-

Notch Gal4 (Bloom 49528) (*Dey et al., 2016*) was crossed to GTRACE stock(Bloom, 28280). F1-Progenies, N > GTRACE, from this cross were used for the experimental analysis. The following strains were a kind gift from Christian Böekel (Technische Universität Dresden, Germany): +;+;Zfh1-T2A-Gal4, tub-Gal80$^{ts}$/TM3 and +;+;Zfh1::GFP/TM3. Crosses were set with +;+;Zfh1-T2A-Gal4, tub-Gal80$^{ts}$/TM3 and GTRACE/Cyo; Gal80$^{ts}$ (Lolitika Mandal, IISCER, Mohali) animals at 18°C. Eclosed animals between 1 to 3 days of age were shifted to 29°C for 72 hr before injury. Injured animals and uninjured controls were incubated at 29°C until dissection.

## Immunohistochemistry and confocal microscopy

Flight muscles were dissected from specifically staged flies were dissected and then fixed in 4% paraformaldehyde diluted in phosphate buffered saline (PBS pH-7.5). For each experiment group in this manuscript, replicates ranged from 6 to 25. Immunostaining was performed according to (*Weitkunat and Schnorrer, 2014*) with few modifications. In brief, samples were then subjected to two washes of 0.3% PTX (PBS + 0.3% Triton-X) and 0.3% PBTX (PBS + 0.3% Triton-X+0.1 %BSA) for 6 hr each. Primary antibody staining was performed for overnight on a shaker and secondary antibodies were added following four washes of 0.3% PTX 2 hr each. Excess of unbound secondary antibodies was removed at the end of 12 hr by two washes of 0.3% PTX 2 hr each following which samples were mounted in Vectashield mounting media. For immunostaining, anti-GFP (Chick, 1:500, Abcam, RRID: AB_300798), anti-Delta (monoclonal mouse, 1:50, Hybridoma bank C594.9B RRID:AB_528194), anti-MHC (Mouse, 1:100, kind gift from Dr. Richard Cripps), TOPRO-3-Iodide (1:1000, Invitrogen), Hoechst 33342 (1:500, ThermoFisher) anti-Neuralized (1:50, Rabbit) (*Lai et al., 2001*), Anti-Zfh-1 (1:1000, rabbit, gift from Prof. Ruth Lehmann lab), Anti-Dmef2 (rabbit 1:3000, gift from Bruce Patterson RRID:AB_2568604) Phalloidin (Alexa-488/647/568 conjugate, 1:500, ThermoFisher), anti-phosphohistone-H3 (Rabbit, 1:100, Millipore), anti-Spectrin (3A9 mouse monoclonal, DSHB 1:5, RRID:AB_528473), anti-Vinculin (1:200, mouse Abcam RRID:AB_444215) antibodies were used. Secondary antibodies (1:500) from Invitrogen conjugated with Alexa fluor-488, 568 and 647 were used in immunostaining procedures.

## Confocal and electron microscopy

For confocal experiments, an Olympus FV 1000 confocal point scanning microscope and Zeiss LSM 780 were used for image acquisition. Images were processed using ImageJ software (Rasband WS, ImageJ U S. National Institutes of Health, Bethesda, Maryland, USA, http://imagej.nih.gov/ij/, 1997–2012). Quantification of number of actively dividing cells in PH-3 labeling experiments was performed as described in *Gunage et al., 2014*).

For electron microscopic analysis, the muscles were processed according to (*Garcia-Murillas et al., 2006*). In brief, flight muscles were dissected in ice-cold fixative (2.5% glutaraldehyde in 0.1 M PIPES buffer at pH 7.4). After 10 hr of fixation at 4°C, samples were washed with 0.1M PIPES, post-fixed in 1% OsO4 (30 min), and stained in 2% uranyl acetate (1 hr). Samples were dehydrated in an ethanol series (50%, 70%, 100%) and embedded in epoxy. Ultrathin sections (50 nm) were cut and viewed on a Tecnai G2 Spirit Bio-TWIN electron microscope. Results presented in this manuscript were replicated eight times.

## Muscle injury

To induce regeneration response in the flight muscle, flies were injured through the cold stab method. Flies were CO2 anesthetized and a single stab injury was performed manually with dissection pin or tungsten needle dipped in liquid nitrogen (Fine Scientific Tools, Item no-26002–10, Minutien Pins-Stainless Steel/0.1 mm Diameter). Care was taken so that the tungsten needle tip did not cross the hemithorax so that the damage was restricted to a minimum. Although completely effective, the method has limited precision. Control flies were age matched adult flies but with no injury to muscles. Injured animals recovered on corn meal *Drosophila* food. The flies were then processed for immunostaining of flight muscles as mentioned in the immunohistochemistry procedure. The detailed protocol of muscle injury can be found at Bio-protocol (*Chakraborty et al., 2018*).

## Statistical analysis

Statistical significance of differences in cell counts between control and treatment groups were calculated by the Student's t test on Microsoft Excel. In all quantitations, the mean and standard error for each group are presented.

## Acknowledgements

This work was possible due to the generous support of the National Centre for Biological Sciences, Tata Institute of Fundamental Research and the J C Bose Fellowship of the Government of India. We thank the Central Imaging and Flow Facilities (NCBS) for the microscopes used in this study. The NCBS Fly facility provided indispensible help with managing fly stocks. The authors would also like to thank Rajan Thakur for all the help provided for electron microscopy related experiments. We are deeply thankful to Ruth Lehmann (NYU, New York, USA) for the Zfh1 antiserum. Special thanks to Christian Böekel (CRT, Dresden, Germany) for Zfh1::Gal4 and Zfh1::GFP lines.

## Additional information

### Competing interests

K VijayRaghavan: Senior editor, *eLife*. The other authors declare that no competing interests exist.

### Funding

| Funder | Grant reference number | Author |
|---|---|---|
| Department of Science and Technology, Ministry of Science and Technology | JC Bose Fellowship | K VijayRaghavan |
| Science and Engineering Research Board | | Dhananjay Chaturvedi |
| Department of Science and Technology, Ministry of Science and Technology | | Rajesh D Gunage |

The funders had no role in study design, data collection and interpretation, or the decision to submit the work for publication.

### Author contributions

Dhananjay Chaturvedi, Conceptualization, Data curation, Formal analysis, Validation, Investigation, Visualization, Methodology, Writing—original draft, Writing—review and editing, Identified Zfh1 as a satellite cell marker and characterized their ability to fuse post injury; Heinrich Reichert, Conceptualization, Formal analysis, Supervision, Methodology, Writing—original draft, Writing—review and editing; Rajesh D Gunage, Conceptualization, Data curation, Formal analysis, Supervision, Validation, Investigation, Visualization, Methodology, Writing—original draft, Writing—review and editing, Identified the muscle lineage of satellite cells and demonstrated their relation to notch signaling; K VijayRaghavan, Conceptualization, Resources, Formal analysis, Supervision, Funding acquisition, Investigation, Writing—original draft, Writing—review and editing

### Author ORCIDs

Dhananjay Chaturvedi (iD) https://orcid.org/0000-0002-3957-1236
Rajesh D Gunage (iD) http://orcid.org/0000-0001-5694-4658
K VijayRaghavan (iD) https://orcid.org/0000-0002-4705-5629

### Decision letter and Author response

Decision letter https://doi.org/10.7554/eLife.30107.020
Author response https://doi.org/10.7554/eLife.30107.021

## Additional files

### Supplementary files

• Transparent reporting form
DOI: https://doi.org/10.7554/eLife.30107.018

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
