## [Decision Letter]

[Editors’ note: a previous version of this study was rejected after peer review, but the authors submitted for reconsideration. The first decision letter after peer review is shown below.]

Thank you for submitting your work entitled "Newly identified satellite cells respond to damage through Notch-Delta signaling to fuse with adult *Drosophila* muscles" for consideration by *eLife*. Your article has been reviewed by three peer reviewers, one of whom is a member of our Board of Reviewing Editors, and the evaluation has been overseen by a Senior Editor. Our decision has been reached after consultation between the reviewers. Based on these discussions and the individual reviews below, we regret to inform you that your work will not be considered further for publication in *eLife*.

All reviewers agree that the identification of satellite cells in *Drosophila* is potentially exciting. However, they feel that the work is preliminary and not well substantiated by the quality of the data presented. Please see reviewers' comments for specific suggestions.

*Reviewer #1:*

In this paper, the authors extended their previous characterization of muscle-specific stem cells in fly development by presenting evidence that the descendants of these stem cells in adult flies similarly as vertebrate muscle satellite cells. This conclusion would be highly significant if substantiated. While the authors have presented some evidence to show that these cells undergo mitosis upon injury, whether their progeny contribute to adult muscle was hard to evaluate, at least based on the GFP data alone. Additional experiments to clearly demarcate these GFP-positive cells as part of the muscle fibres would strengthen the conclusion.

Reviewer #2:

In their paper the authors attempt to extend their previous findings published in *eLife* that a set of cells known as AMPs, which give rise to muscle during early development, are also capable of acting as satellite cells in the adult.

As the paper currently stands the data presented are not convincing. No evidence presented for regeneration, let alone regeneration of new tissue that leads to functional tissue. Without more rigorous data, in particular better clonal data, I would not recommend publication. In fact, at this point the data, at best, suggest that the AMP remnants stick around and can divide but do not give rise to any new cells, as has been shown to be the case by Fox and Spradling with the adult hindgut imaginal ring. Below are experiments that would go a long way to proving the author's points. I would strongly encourage the authors to address most of the comments with new, better, and more rigorous experiments rather than arguing them away. Ultimately, either there are or there aren't satellite cells that give rise to new muscle following injury AND are necessary for regeneration. And a publication claiming there are satellite stem cells based on poor quality data, which is later shown to be incorrect would reflect poorly on the authors and *eLife*.

Major issues:

1) How many satellite cells do the authors think there are? What is the ratio between satellite and differentiated cells?

2) Figure 1: Without better markers I cannot tell where the GFP labeled cells are.

3) The EdU data, including methodology, should be included.

4) How close is panel C to the big hole shown in A?

5) Why is the GFP so weak/strange looking in E in comparison to Figure 1D-E? And the merge in H? There is a lot of GFP that is of equal or more intense than said clones and not associated with anything.

6) Figure 3D: Over what area and what percentage of satellite cells? How was the region chosen? Do all satellite cells become mitotically active following injury? 50%? Also, I'm surprised the difference here is only 2 stars.

7) Why are most of the GFP descendents PH3 positive 24 hours after injury? I would expect only half to be positive since half of the progeny would not be satellite cells, but daughters of satellite cells. Do the authors think the daughters can divide? Or perhaps do the authors think the response by satellite cells takes 24 hours? Does PH3 number go down with time? If so what are the kinetics? Knowing this would make testable predictions about when one should see labeled cells outside and inside the muscle.

8) The quality of the PH3 staining is poor and therefore not convincing. It essentially looks like background. The presence of mitotic spindles would help the author's case. Along these lines what does the equivalent of 3E-3H PH3 staining look like in uninjured tissue? They could help with background green levels.

9) Can the authors rule out dividing cells as being hemocytes?

10) Figure 3I-3N: Why do the authors induce clones 6hr prior to injury whereas in earlier experiments they induced clones in the larval period?

11) Figure 3I-3N: Nothing in this figure leads me to believe that there are now GFP+ cells in the muscle. The position of the nucleus appearing to "intercalate" could be based solely on optical sectioning. The satellite cells should both self-renew and give rise to a myoblast that differentiate and fuse, so there should always be two labeled cells, right? Where is the second cell? Better markers, close–up views, and 3-D reconstruction would bolster the author's assertion. Furthermore, I would like to get a quantification of how many new muscle nuclei there are.

12) Ultimately, the authors need to use a reporter that would be capable of labeling either the muscle membrane or cytoplasm so we can see the new cell and the extent of repair. I understand the author's argue that the CD8::GFP gets diluted. But with enough damage it should be possible for enough new nuclei to be made to produce an identifiable signal. Or to use more sensitive membrane reporter, for example see "ultrafast tissue staining with chemical tags" published in PNAS by Kohl et al., (2014).

13) The presence of a muscle fibre that is GFP positive cannot rule out fusion of a "satellite cell" directly with a fibre versus the progeny of a "satellite cell" fusing with a muscle fibre. The use of twin spot clonal systems (those in which each progeny is labeled with a different color following mitotic recombination) could help resolve this issue.

14) Notch staining in 4A-C looks strongly nuclear. In fact, Notch staining in flies tends to be membrane and in endocytic vesicles and never seen in the nucleus. In addition, there is no control using N RNAi to show staining goes away. This is true of Delta and neutralized with respect to lack of controls for immune-reactivity.

15) Figure 4N: I assume the area examined is the same for PH3 experiments. If so why are 5% of cells NRE+. In other words, there are approx. 175 dividing cells in Figure 4D but 10-12 NRE positive cells in Figure 4N.

16) Figure 4O-P: Where is Delta? Is it nuclear again, like N staining? Should be either membrane or vesicular. These low magnification images are not informative at all. Also, it appears that Dl staining is present in the Delta-RNAi image?

17) The neuralized staining looks like background and could benefit from a control (neutralized knockdown followed by staining). Also – are the levels adjusted the same? R inset looks like it has less background than S inset.

18) Does neur knockdown in muscle (using Act88F) also block PH3? What happens with Notch knockdown using Act88F?

Reviewer #3:

The article by Gunage et al. describes a MARCM clonal analysis in *Drosophila* aimed at identifying adult muscle satellite cells. That is, non-differentiated myoblasts associated with the mature adult muscles that might contribute to adult muscle repair upon activation. The significance of this finding, if proven, is extremely high: there has been no prior evidence that satellite cells exist in *Drosophila*, yet the existence of such cells would enable the use of a powerful system to investigate the specification and biology of these cells. Such a finding would be a major advance.

In their paper, Gunage et al. identify clones of cells expressing dMEF2-Gal4 that appear to satisfy some of the criteria of satellite cells. They are located at or near the adult flight muscles; there is some evidence that they proliferate in response to injury; and aspects of the biology of these cells is affected by manipulation of the Notch pathway, that regulates the specification of *Drosophila* adult myoblasts, and plays a major role in satellite cell activation in vertebrates.

Despite these promising findings, I feel that the depth of analysis presented here does not go far enough to confirm the existence of these cells in *Drosophila*. Given the significance of the proposed conclusions, I think that the analyses should be far more rigorous than shown here. Specific concerns are as follows:

1) The stains and documentation in numerous figures do not effectively support the authors claims about the cells. For example, in Figure 1 panels C and D, it is not apparent that the GFP-labeled cells are located at the periphery of the muscle, because there are no markers used to label the periphery. Also, the red dotted lines are in slightly different locations comparing panel D with E, further raising doubt as to the precise location of the boundaries of the muscle cells. I suggest co-staining with a membrane marker to more clearly localize the cells. In addition, Figure 1 panels F and G are so weakly stained that it is not possible to see what is intended.

2) Figure 2 shows electron micrographs of flight muscle cells and a cell associated with the flight muscles. There is no evidence that this cell either corresponds to a satellite cell or shares any of the markers of satellite cells. I think this figure should be removed.

3) Since dMEF2-Gal4 is expressed in cells other than adult muscle precursor cells associated with the wing discs, the authors cannot claim that the GFP-positive cells observed in the adult arise from the wing disc myoblasts without more detailed lineage tracing carried out during the pupal stage.

4) There are no markers that are specific for the cells identified as satellite cells. An analysis of a number of candidate markers would improve confidence that the identified cells are myoblast-like, such as Twist, dMEF2, and others, and would improve the ability of the authors to follow the cells as they proliferate. Otherwise, one might argue that the infiltrating cells are non-muscle cells, perhaps responding to injury or sepsis, and unrelated to a repair mechanism.

5) Figure 3. It is difficult to reconcile some of the data presented in this figure. Firstly, can the authors describe (or ideally show) the proximity of the stained areas to the muscle injury? This applies to all subsequent figures where the muscle is injured. It would be good to confirm that the band of PH3-positive cells is close to the site of damage, and does not occur at a location away from the area of damage. Otherwise, the PH3 stain might instead represent some kind of cellular response. In panels C and F there appear to be a relatively large number of PH3-positive cells, whereas in panels I-N the number of cells is quite sparse. While I understand that the GFP marking system in I-N only labels a subset of cells, it would be nice to know how many such satellite cells exist, and whether the number of marked cells in C and F is consistent with the numbers of precursors. This also relates to comment 4 above, where there is a paucity of markers for these important cells. Also, the PH3 and GFP stains in panels E and F are too faint to be considered reliable. Finally, in panels J-N, the authors show GFP cells associated with an injury site, but the location of the injury is not shown, and the relative locations of the GFP-positive cells relative to the cell membrane cannot be determined, thus it is not clear if the cells are really infiltrating.

6) Figure 4. The Notch staining in panel B is not convincing. Why is the DNA stain in panels H-J fainter than the same stain in panels K-M?

7) For the Delta knockdown experiment and for the Neuralized expression levels, there must be quantitation of the degree of knockdown and the degree of Neur over-expression. Comparison of stained sections is not satisfactory, and instead the authors should carry out quantitative RT-PCR or western blotting to confirm their stains. Also, why is the Phallodin intensity greater in panel S than R? Accurate quantitation as suggested here would protect the authors from concerns that differential efficiency of staining or imaging is the cause of the observed changes in intensity.

8) There is no evidence that the adult muscles repair, that the identified cells have any role in the repair, or that the repair depends upon Notch pathway members.

[Editors’ note: what now follows is the decision letter after the authors submitted for further consideration.]

Thank you for submitting your work entitled "Identification and Functional Characterization of Muscle Satellite Cells in *Drosophila*" for consideration by *eLife*. Your article has been reviewed by two peer reviewers, and the evaluation has been overseen by a Reviewing Editor and a Senior Editor. The reviewers have opted to remain anonymous.

Our decision has been reached after consultation between the reviewers. Based on these discussions and the individual reviews below, we regret to inform you that your work will not be considered further for publication in *eLife*.

While both reviewers appreciate the improvement over the last submission, they remain unconvinced that the study definitively shows the origin, existence and activity of these cells similar to the mammalian satellite cells. They are especially concerned about the quality of the lineage and functional data implicating these cells in muscle repair.

Reviewer #1:

In their resubmission, the authors argue that *Drosophila* flight muscles contain satellite cells that proliferate in response to damage and give rise to progeny that fuse with muscle to lead to functional repair. While I agree that there has been an improvement in the quality of data since their last submission, in particular the presence of superficial cells, I am not convinced that these cells are a stem cell population that participates in muscle repair. The lineage and functional data presented is poor quality, not only from the level of cellular resolution but with respect to the assays used. I think part of the problem is the syncytial nature of the tissue, which makes it hard to analyze. Definitive proof will probably require live imaging in intact flies, something that I would thing feasible with 2 photon microscopy.

Major issues:

1) Are the muscle nuclei polyploid? Are the green arrow nuclei diploid? If so it would help to DAPI quantify.

2) In Figure 1A there is only one superficial nuclei. In 1C there are three. The scale bar suggests the picture is the same size but 1C looks smaller?

3) "(In control experiments, MARCM clones were also triggered in pupal stages and recovered in the adult to confirm that the labeled cells were not infiltrating cells that might derive from unknown proliferating cells located external to the wing disc; see Figure 3—figure supplement 1A-C)”. I don't understand this. Is the reasoning that during pupation there are no cell divisions in this lineage and therefore you would expect not to recover clones? Why is it parentheses? Also, throughout the paper there are examples of text placed in parentheses. Is this a convention the authors use to indicate an afterthought? It's unnecessary.

4) "Note that, while differentiated myoblasts are also targeted in this MARCM experiment, no GFP labeled cells are visible within the adult muscle fibres since their membrane tethered GFP becomes diffuse due to incorporation into the extensive muscle fibre membrane following cell fusion (see schematic in Figure 3D)." Wouldn't the argument used by the authors later, that gal80 is present, explain why there is no membrane GFP?

5) "Care was taken to restrict damage such that only 1 or 2 muscle fibres were affected and that fibres were not severed by the injury (Figure 4A, B)." Out of how many fibres? How much area would this represent?

6) "DLMs damaged in this way can regenerate and in morphological respects appear normal after approximately 3 weeks (Figure 4—figure supplement 1)."

7) How do they appear after 1 day? 2 days? 3 days? etc. It's confusing why one would look after 3 weeks, especially when data presented later implies that muscle function returns after 2 days! By looking at many early time points one might get an idea of how the proposed repair occurs. In fact, maybe the area of damage is repaired by fusion of remaining tissue or hypertrophy of adjacent nuclei. There is no way to rule these alternatives out based on the resolution (spatial and temporally) provided. Also, it's unclear where the original injury might have been. There is no presentation of muscles that have no/poor repair (e.g. DmefGal4 X Notch RNAi) to compare with.

8) "For both PH-3 and EdU labeling, evidence for increased mitotic activity was largely restricted to the damaged muscle fibres and rarely observed in undamaged muscle fibres”. Could you show a damaged fibre next an undamaged fibre with PH3 staining.

9) In contrast, within the damaged muscle fibre, evidence for increased mitotic activity was seen in nuclei along the entire extent of the fibre length. It surprises me that mitosis would be seen along the entire fibre. How many nuclei would that be? Is that the case with vertebrate satellite muscle cells? How many of new progeny fuse with the fibre?

10) "In order to label only cells that were generated by mitotic activity in the adult, clones were induced in the adult 6h prior to physical injury and recovered 24h later." According to the authors divisions are rare, so how were clones recovered at all? The better way is follows right after in the parentheses.

11) "This implies that some of the daughter cells generated by satellite cells during injury-induced proliferative activity remained at the muscle cell surface while others appear to have entered muscle fibre's interior and may have fused with the injured muscle cell." This was not convincing. They may have not entered the interior but rather intermixed in the space which has been disrupted by injury. Or that injury leads to non-specific fusion caused by injury.

12) The FUCCI system does not provide information about lineage. And the quantity and quality of the data was not better than PH3. Along these lines, could the authors show examples of metaphase, anaphase and telophase? This would be much more convincing that the surprisingly numerous PH3 positive cells. Also, the long chain of "proliferative" cells shown using the FUCCI system was never seen or described in earlier experiments and makes me wonder if it's an artifact.

13) Nts affects the entire animal. DmefGal4 X Notch RNAi affects all muscle. MARCM clones of Notch mutants would be more useful since one could see they are PH3 negative in a sea of PH3 positive WT cells.

14) Was DAPT fed to the flies, for how long?

15) The Notch responsive element is not a GFP construct of E(spl). It's, I believe, Su(H) binding sites fused to Gbe binding sites and based on the original reporter made by Sarah Bray's lab.

16) "Targeted Notch downregulation led to significant perturbation in the flight initiation response, indicating the importance of satellite cell lineage proliferation and fusion to restore muscle function after injury." Actually, this is merely a strict correlation. That is, PH3 falls and so does flight initiation. But whether they are directly connected cannot be concluded. Notch may be required in Dmef2 positive cells for other things. I am also concerned and surprised that repair occurs in 2 days. And along these lines why in Figure 4—figure supplement 1 did the authors wait until 3 weeks for evidence of repair? Furthermore, what about Notch RNAi after 4 days, 5 days? (They only look at Notch RNAi after 2 days). Given the huge variation even among each genotype it's hard to see how one can conclude statistical significance. Can it recover? I would not expect it to given how well PH3 levels fall. Therefore, if it recovered it would argue something else is going on. And why only do 3 trials? Given the ease of the experiment you could do 300 trials.

17) It's unclear why the authors chose a red box next to the area of damage as opposed to a box with the area of damage in the middle. And why a 100uM and not more? At what distance does PH3 fall? Or is it an entire fibre becomes activated?

*Reviewer #2:*

The manuscript by Gunage et al. is a significantly improved version of a manuscript submitted earlier this year, that attempts to describe and characterize for the first time satellite cells in *Drosophila*. This appears a particularly difficult task, given a paucity of suitable markers for these cells and no established muscle injury model in this organism. Nevertheless, the significance of this work, if proven, would be high.

The prior version of the manuscript suffered from a lack of suitable counterstains and marker analysis for the satellite cells, and this has been improved. It is clear that a separate population of cells exists on the flight muscle surface, and that upon injury there is an increase in the number of proliferating cells in and around the flight muscles. However, I still remain a little unconvinced that this revised work still definitively shows the origin, existence and activity of these cells.

Firstly, there is still no good data supporting that the satellite cells arise from the wing imaginal discs, versus any other discs or any other part of the body that expresses Dmef2-gal4. The authors indicate they studied pupal time points in a supplementary figure, however that data is poorly described and does not appear to represent a time course.

Secondly, the authors report that they were not able to identify any markers of the satellite cells from a candidate list. It is unclear if they have tested Dmef2 itself. The satellite cells should be positive for this protein because the Dmef2-gal4 driver is active in the cells.

Thirdly, the authors are still struggling to definitively demonstrate that the satellite cells and their daughter cells correspond to the nuclei that are thought to infiltrate the injured muscle. In Figure 4D, for example, it looks as though essentially every nucleus becomes PH3 positive. Do the endogenous muscle nuclei (i.e. those that were present in the muscle prior to injury) also accumulate PH3? A clarification of not only the number of PH3-poistive nuclei in each sample, but also the total number of nuclei, would help resolve this. This point is not well supported by the data in Figure 4 H-K, where the PH3 stain is so weak it could be argued to be either present in all nuclei or absent from all nuclei.

In addition, the notion that some of the activated satellite cells remain on the surface of the muscle is attractive, but is not well supported by the existing data. Here, there is no proof that the surface nuclei are not fused to the muscle (and no support from the data in panel 4S-U), nor that the activated cells do not fuse to the muscle in the following day or so. Later time points following injury and MARCM induction might resolve this issue.

Finally, it is still not clear what is the level of repair taking place. The rescue of flight defects is modest and not necessarily due to muscle defects (it could be some form of shock or inflammatory response that causes a reduction in flight after injury); the authors do not provide any data suggesting that new myofibrillar proteins are being made; and there is no indication that there is an overall increase in the number of muscle nuclei following repair. Thus, the roles of these cells are not defined.

[Editors’ note: what follows is the decision letter after the authors resubmitted for further consideration.]

Thank you for submitting your article "Identification and Functional Characterization of Muscle Satellite Cells in *Drosophila*" for consideration by *eLife*. I do apologise for the very long time it took to review your manuscript. The senior editors who saw the original version have stepped down as editors, and none of the three in-depth reviewers was available to comment. This, compounded with your August submission, when many editors were on vacation, explains – but of course does not excuse – the delay. Your article has been reviewed by a single peer reviewer, and the evaluation has been overseen by Fiona Watt as the Senior Editor. The reviewer opted to remain anonymous.

Our normal practice is to have a discussion between multiple reviewers to achieve a consensus position on each manuscript. However, in this case there was only one in-depth review and so we have provided it in full below. Both the reviewer and the editor are supportive of publication, provided that you address each of the points raised.

Reviewer #1:

The manuscript by Gunage et al. provides an analysis of adult flight muscles in *Drosophila* and presents evidence consistent with the existence of cells within these muscles that possess properties that are highly similar to those that have been described for mammalian satellite cells. These properties include localization at the periphery of muscle fibres, derivation from developmentally active muscle precursors, proliferation in response to muscle damage, contribution of nuclei into regenerated muscle fibres, and dependence on Notch signaling for myogenic activation. The authors also identify a new marker of these cells Zfh-1 and establish a new muscle injury/repair model in the fly. While much of the data presented are descriptive in nature, the clonal analyses and lineage tracing are compelling in support of the authors' hypothesis. The manuscript thus represents an important first step, and although clearly there is much more work to be done to fully understand the nature and activities of these cells this work is likely to impact the field by advancing the relevance of the fly system for studies of muscle regenerative biology and providing a new model system for genetic and mechanistic studies. For these reasons, I am supportive of publication in *eLife*, assuming the authors can address the few points below:

1) Subsection “Two different types of cells are present in adult flight muscle”: "This observation is confirmed by co-staining these adult muscle fibres for expression of either Act88F, an indirect flight muscle specific isoform of actin, or Tropomoysin." Is this data contained in the manuscript? If so, it should be properly called out and if not, it should be added.

2) The authors should provide in the Discussion section more information about their newly defined marker Zfh-1, including its other known functions in the fly (it appears to be involved in muscle development and in self-renewal of germline stem cells) as well as information about whether its homolog is expressed in mammalian satellite cells (for which multiple publicly available gene expression data sets are available).

3) Statistical tests used should be identified in the figure legends and methods should describe statistical approaches and randomization (if used).

4) If technically feasible, immunoEM to demonstrate that the "satellite cells" identified by ultrastructural analysis are the same cells as those identified by confocal microscopy and Zfh-1 expression would greatly enhance the authors' conclusions.

---

## [Author Response]

[Editors’ note: the author responses to the first round of peer review follow.]

Reviewer #1:In this paper, the authors extended their previous characterization of muscle-specific stem cells in fly development by presenting evidence that the descendants of these stem cells in adult flies similarly as vertebrate muscle satellite cells. This conclusion would be highly significant if substantiated. While the authors have presented some evidence to show that these cells undergo mitosis upon injury, whether their progeny contribute to adult muscle was hard to evaluate, at least based on the GFP data alone. Additional experiments to clearly demarcate these GFP-positive cells as part of the muscle fibres would strengthen the conclusion.

We have addressed above concerns by several additional experiments using methods such as EdU labeling, Fly-FUCCI and 3D reconstruction. Further, the role of these cells and of the Notch pathway in repair after injury is now strengthened behavioral assays.

Reviewer #2:In their paper the authors attempt to extend their previous findings published in eLife that a set of cells known as AMPs, which give rise to muscle during early development, are also capable of acting as satellite cells in the adult.As the paper currently stands the data presented are not convincing. No evidence presented for regeneration, let alone regeneration of new tissue that leads to functional tissue. Without more rigorous data, in particular better clonal data, I would not recommend publication. In fact, at this point the data, at best, suggest that the AMP remnants stick around and can divide but do not give rise to any new cells, as has been shown to be the case by Fox and Spradling with the adult hindgut imaginal ring. Below are experiments that would go a long way to proving the author's points. I would strongly encourage the authors to address most of the comments with new, better, and more rigorous experiments rather than arguing them away. Ultimately, either there are or there aren't satellite cells that give rise to new muscle following injury AND are necessary for regeneration. And a publication claiming there are satellite stem cells based on poor quality data, which is later shown to be incorrect would reflect poorly on the authors and eLife.Major issues:1) How many satellite cells do the authors think there are? What is the ratio between satellite and differentiated cells?

We have counted the numbers of satellite cells and differentiated cells in the DLMs. There are 20 ± 4 satellite cells for 700 ± 50 differentiated cells for one (of 6) DLMs. This has been incorporated in the text.

2) Figure 1: Without better markers I cannot tell where the GFP labeled cells are.

In addition to the experiments described in this figure with superimposed stacks, we have carried out new labeling experiments using different markers. We present this labeling of cells in a new figure panel, which also has a scan in the z axis that allows the reader a 3D view of the labeled cells in relation to the muscle fibres. In addition (Figure 1—figure supplement 1) we show that these cells do not express flight muscle-specific Actin88F or Tropomyosin.

3) The EdU data, including methodology, should be included.

We present the EdU data in a new figure panel (Figure 4F, G). We also include the methodology in a new paragraph of the Materials and methods section.

4) How close is panel C to the big hole shown in A?

We mention that the panel C is at about 100 micrometers from the site of injury in the corresponding Results section (Figure 4A).

5) Why is the GFP so weak/strange looking in E in comparison to Figure 1D-E? And the merge in H? There is a lot of GFP that is of equal or more intense than said clones and not associated with anything.

We have repeated the corresponding experiments and now present new panels corresponding to Figure 4H-K which show unfused GFP labeled cells more clearly with much less background (which is likely to be due to fusion of labeled cells). The difference in GFP staining in these panels to that shown in Figure 3E-G is less in the case illustrated, although differences are to be expected since the clone induction time is different (late third instar versus adult) for wild type (Figure 3E-G) compared to damaged muscle (Figure 4H-K).

6) Figure 3D: Over what area and what percentage of satellite cells? How was the region chosen? Do all satellite cells become mitotically active following injury? 50%? Also, I'm surprised the difference here is only 2 stars.

The area being considered is now shown in the panel in low magnification of the injured muscle in Figure 4A shown as a red dotted rectangle (a corresponding area was taken from uninjured control). We have redone the experiment and analysis and corresponding results are now part of Figure 4.

7) Why are most of the GFP descendents PH3 positive 24 hours after injury? I would expect only half to be positive since half of the progeny would not be satellite cells, but daughters of satellite cells. Do the authors think the daughters can divide? Or perhaps do the authors think the response by satellite cells takes 24 hours? Does PH3 number go down with time? If so what are the kinetics? Knowing this would make testable predictions about when one should see labeled cells outside and inside the muscle.

New experiments added, now Figure 4H-K (magnified inset in Figure 4K) clearly shows example of lineage trace and PH3 labeling demonstrating mitotically active as well as inactive satellite cells. Our results based on PH3, Edu labeling (Short pulse and chase) and Fly-FUCCI show less than 50% of unfused cells are mitotically active.

For these experiments clonal labeling was performed during larval stages and clones were recovered post injury 12h. Mitotic activity of activated stem cells decreases with time after injury and our analysis of clonal analysis has clearly revealed the fusion is active around 24h as shown in Figure 4L-Q and Figure 4S-U.

8) The quality of the PH3 staining is poor and therefore not convincing. It essentially looks like background. The presence of mitotic spindles would help the author's case. Along these lines what does the equivalent of 3E-3H PH3 staining look like in uninjured tissue? They could help with background green levels.

We have repeated the corresponding out experiments and provided new figures (Figure 4H-K and also Figure 4—figure supplement 2A, B) in which the staining quality is substantially improved. Also, we have used Fly-FUCCI to demonstrate the same and the results are now included in the Figure 4—figure supplement 2C, D.

9) Can the authors rule out dividing cells as being hemocytes?

To rule this out, we have carried out labeling experiments with the e33c-Gal4 line that marks hemocytes (Fossett et al., 2003, Matova and Anderson, 2006). The dividing cells were not labeled indicating that they are not hemocytes. This experiment is now incorporated into the Results section in Figure 3—figure supplement 1E, F.

10) Figure 3I-3N: Why do the authors induce clones 6hr prior to injury whereas in earlier experiments they induced clones in the larval period?

Clones were induced in the adult and not in the larva in order to label only those lineages that had proliferated specifically in the adult in response to injury and not those lineages, which had proliferated earlier (Figure 3I-3N of earlier MS is now become Figure 4L-Q). The 6 h time period of clone induction prior to injury was chosen to ensure that adequate FLP-ase protein is present at the time of injury. We now mention this in the Results section to clarify this.

11) Figure 3I-3N: Nothing in this figure leads me to believe that there are now GFP+ cells in the muscle. The position of the nucleus appearing to "intercalate" could be based solely on optical sectioning. The satellite cells should both self-renew and give rise to a myoblast that differentiate and fuse, so there should always be two labeled cells, right? Where is the second cell? Better markers, close up views, and 3-D reconstruction would bolster the author's assertion. Furthermore, I would like to get a quantification of how many new muscle nuclei there are.

We have carried out a 3-D reconstruction and present this together with close–up view and a visualization of all optical sections in a movie in new figure panels (Video 5 and Video 6, 7); this clearly shows that there are GFP positive cells in the muscle. We have also carried out a quantification of how many new muscle nuclei there are in induced clones and show this in a new figure panel (Figure 5O). While we understand the logic of the referee in assuming that there should always be two labeled cells, we cannot assert that this must the case for two reasons. First, we have not studied the mode of cellular proliferation of the satellite cell lineages following injury and thus cannot say if it is asymmetric or symmetric and if it includes transit- amplifying cells. Second, given the amount of time between clone induction and recovery, the satellite cells could divide more than once thus producing more than one round progeny (even if there are no transit-amplifying cells). Our results with Fly-FUCCI experiments showed a clear lineage and hints towards possible mechanisms referees are suggesting (Figure 4—figure supplement 2C, D). As we mention in the Discussion section, a detailed characterization of the proliferation properties of satellite cells in response to injury is an important topic for further investigation.

12) Ultimately, the authors need to use a reporter that would be capable of labeling either the muscle membrane or cytoplasm so we can see the new cell and the extent of repair. I understand the author's argue that the CD8::GFP gets diluted. But with enough damage it should be possible for enough new nuclei to be made to produce an identifiable signal. Or to use more sensitive membrane reporter, for example see "ultrafast tissue staining with chemical tags" published in PNAS by Kohl et al., (2014).

We have repeated these experiments and the results are included in the Figure 4S-U.

13) The presence of a muscle fibre that is GFP positive cannot rule out fusion of a "satellite cell" directly with a fibre versus the progeny of a "satellite cell" fusing with a muscle fibre. The use of twin spot clonal systems (those in which each progeny is labeled with a different color following mitotic recombination) could help resolve this issue.

We cannot claim that only the satellite cell progeny but not the satellite cell can fuse directly with a muscle fibre. It is indeed possible that both occur. Our experiments show that following the injury induced proliferation, some of the cells in the satellite cell lineage fuse with the muscle fibres and other do not (Figure 4F-R). We have now mentioned this explicitly in the results (Figure 4—figure supplement 2C, D) and Discussion section.

14) Notch staining in 4A-C looks strongly nuclear. In fact, Notch staining in flies tends to be membrane and in endocytic vesicles and never seen in the nucleus. In addition, there is no control using N RNAi to show staining goes away. This is true of Delta and neutralized with respect to lack of controls for immune-reactivity.

The referee is correct in that staining for the N intracellular domain is definitely not nuclear. The impression from the figure mentioned, that it is nuclear is due to the fact that the nucleus of the satellite cell is relatively large compared to the rest of the cytoplasm (see our EM figure). To show that N intracellular domain is not in the nucleus, we have carried out high resolution imaging of the satellite cells co-labeled for N intracellular, cell membrane and nucleus. We present this in a new figure panel (Figure 5A-D). We have also carried out the control using N RNAi; this is presented in a new figure panel. Similarly, we have carried out the controls for Delta (Figure 6 A, B) and Neuralised (Figure 6 F, G); this is presented in new figure panels.

15) Figure 4N: I assume the area examined is the same for PH3 experiments. If so why are 5% of cells NRE+. In other words, there are approx. 175 dividing cells in Figure 4D but 10-12 NRE positive cells in Figure 4N.

The area examined is the same, and the numbers are as presented. Thus, under control conditions we count 175 dividing cells and only around 10 NRE positive cells. We do not claim to investigate the mechanisms responsible for this difference in numbers, rather we simply show that in both cases (division and NRE expression) there is a marked increase in the number of implicated cells following injury.

16) Figure 4O-P: Where is Delta? Is it nuclear again, like N staining? Should be either membrane or vesicular. These low magnification images are not informative at all. Also, it appears that Dl staining is present in the Delta-RNAi image?

We have carried out new experiments and present new magnifications that clearly show that Delta is not nuclear and that it is not present in the RNAi image. These data are shown in new figure panels (Figure 6A, B).

17) The neuralized staining looks like background and could benefit from a control (neutralized knockdown followed by staining). Also – are the levels adjusted the same? R inset looks like it has less background than S inset.

We have carried out new experiments including controls that now clearly document the neuralized staining. These data are shown in new figure panels (Figure 6F, G).

18) Does neur knockdown in muscle (using Act88F) also block PH3? What happens with Notch knockdown using Act88F?

We have carried out new experiments that show that both Neur and N knockdown downregulate proliferation. These data are shown in new figure panels (Figure 6 IK).

Reviewer #3:[…] Specific concerns are as follows:1) The stains and documentation in numerous figures do not effectively support the authors claims about the cells. For example, in Figure 1 panels C and D, it is not apparent that the GFP-labeled cells are located at the periphery of the muscle, because there are no markers used to label the periphery. Also, the red dotted lines are in slightly different locations comparing panel D with E, further raising doubt as to the precise location of the boundaries of the muscle cells. I suggest co-staining with a membrane marker to more clearly localize the cells. In addition, Figure 1 panels F and G are so weakly stained that it is not possible to see what is intended.

We have repeated the corresponding experiments and now present new panels which show the unfused GFP labeled cells more clearly with much less background (Figure 1C-D, Figure 3—figure supplement 1). In addition, we have carried out new labeling experiments using different markers including a membrane marker. We present this labeling of cells in new figure panels, which also has a scan in the z axis that allows the reader a 3D view of the labeled cells in relation to the muscle fibres (3D video showing two types of cells see Video1).

2) Figure 2 shows electron micrographs of flight muscle cells and a cell associated with the flight muscles. There is no evidence that this cell either corresponds to a satellite cell or shares any of the markers of satellite cells. I think this figure should be removed.

We include this micrograph to show unequivocally that there are small cells in the adult muscle that are not fused with the muscle fibres but are very closely associated with muscle and even share the muscle cell ECM (previously shown in Figure 1D-F, 3D reconstruction Video 4, 5 and also demonstrated in Figure 3—figure supplement 1). Cells of this morphological type are novel and have not been previously described. This clearly confirms the notion that there are two different types of cells in adult muscle, one type has fused its nucleus with the muscle cell and the other novel type, while closely associated with the muscle, has not. We clarify this in the revised Results section.

3) Since dMEF2-Gal4 is expressed in cells other than adult muscle precursor cells associated with the wing discs, the authors cannot claim that the GFP-positive cells observed in the adult arise from the wing disc myoblasts without more detailed lineage tracing carried out during the pupal stage.

We have traced the dMEF2-Gal4 expressing cells associated with the larval wing discs through pupal stages and into the adult to confirm that the labeled cells in the adult do arise from the wing disc muscle precursors. Representative staining results from pupal stages are shown in a new figure (Figure 3—figure supplement A-C).

4) There are no markers that are specific for the cells identified as satellite cells. An analysis of a number of candidate markers would improve confidence that the identified cells are myoblast-like, such as Twist, dMEF2, and others, and would improve the ability of the authors to follow the cells as they proliferate. Otherwise, one might argue that the infiltrating cells are non-muscle cells, perhaps responding to injury or sepsis, and unrelated to a repair mechanism.

We have screened for other candidate markers, but have not found any that are specific for these satellite cells. (This could be a reason why these cells have not been identified previously in *Drosophila*). To rule out that these cells might be infiltrating hemocytes, we have carried out labeling experiments with the Gal4e33 line which marks hemocytes. The dividing cells were not labeled indicating that they are not hemocytes. This experiment is now incorporated into the Results section Figure 3—figure supplement 1E, F.

5) Figure 3. It is difficult to reconcile some of the data presented in this figure. Firstly, can the authors describe (or ideally show) the proximity of the stained areas to the muscle injury? This applies to all subsequent figures where the muscle is injured. It would be good to confirm that the band of PH3-positive cells is close to the site of damage, and does not occur at a location away from the area of damage. Otherwise, the PH3 stain might instead represent some kind of cellular response.

We now show the site of the labeled cells (Figure 4A shown in red dotted rectangle) relative to the site of injury in a new panel, which presents low magnification of the injured muscle (Figure 4A white dotted circle).

In panels C and F there appear to be a relatively large number of PH3-positive cells, whereas in panels I-N the number of cells is quite sparse. While I understand that the GFP marking system in I-N only labels a subset of cells, it would be nice to know how many such satellite cells exist, and whether the number of marked cells in C and F is consistent with the numbers of precursors. This also relates to comment 4 above, where there is a paucity of markers for these important cells.

We have determined the number of unfused cells (mentioned in main text result) and have also determined that 50% percent of the unfused cells are Ph3 positive, indicating that less than 50% of the satellite cells have become active following injury (Mitotic activity is seen primarily in injured area). This is now mentioned in the corresponding Results section (Figure 4C-E and H-K). We have also carried out a quantification of how many new muscle nuclei there are in induced clones and show this in a new figure panel (Figure 5O).

Also, the PH3 and GFP stains in panels E and F are too faint to be considered reliable. Finally, in panels J-N, the authors show GFP cells associated with an injury site, but the location of the injury is not shown, and the relative locations of the GFP-positive cells relative to the cell membrane cannot be determined, thus it is not clear if the cells are really infiltrating.

We have repeated the corresponding out experiments and provided new figures in which the staining quality is substantially improved. We provide new figure panels in which the location of the injury relative to the stained cells is shown (see above). We have carried out a 3-D reconstruction and present this together with close–up view and a visualization of all optical sections in a movie in new figure panels; this clearly shows that there are GFP positive cells in the muscle. Moreover, in new experiments (Figure 4S-U) we show fused membrane GFP cells (white dotted circle) in the vicinity of unfused GFP positive cells (red arrow). These findings are presented in new figure panels incorporated into the Results section.

6) Figure 4. The Notch staining in panel B is not convincing. Why is the DNA stain in panels H-J fainter than the same stain in panels K-M?

We have carried out new labeling experiments and present new magnifications that more clearly show the Notch (and Delta) staining. These data are shown in Figure 5A-D and Figure 6A, B.

7) For the Delta knockdown experiment and for the Neuralized expression levels, there must be quantitation of the degree of knockdown and the degree of Neur over-expression. Comparison of stained sections is not satisfactory, and instead the authors should carry out quantitative RT-PCR or western blotting to confirm their stains. Also, why is the Phallodin intensity greater in panel S than R? Accurate quantitation as suggested here would protect the authors from concerns that differential efficiency of staining or imaging is the cause of the observed changes in intensity.

We have repeated all the experiments and the new panels are included in the result section Figure 6B, H and Figure 6—figure supplement 1 and explained in the Materials and methods section.

8) There is no evidence that the adult muscles repair, that the identified cells have any role in the repair, or that the repair depends upon Notch pathway members.

We present new data showing that adult muscles do repair in anatomical terms, and that this repair depends on the N pathway in a new figure. To address the role of Notch pathway in repair, a functional assay is now included in the Figure 5 panel W, X and Figure 4—figure supplement1.

[Editors' note: the author responses to the second round of review follow.]

We thank both reviewers for their very useful comments, all of which we have addressed in the extensively revised manuscript. We request that this be considered a new submission as in the revised manuscript virtually all of the figures have been replaced by new experiments and figures and text which, we feel have substantially improved the manuscript.

Central among these are new experiments that identify and utilize Zfh1 as a highly specific marker for satellite cells in adult muscle. With this marker we also perform new lineage tracing experiments to confirm the origin of satellite cells and demonstrate that their lineal progeny fuse with muscle fibers after injury.

Reviewer #1:In their resubmission, the authors argue that Drosophila flight muscles contain satellite cells that proliferate in response to damage and give rise to progeny that fuse with muscle to lead to functional repair. While I agree that there has been an improvement in the quality of data since their last submission, in particular the presence of superficial cells, I am not convinced that these cells are a stem cell population that participates in muscle repair. The lineage and functional data presented is poor quality, not only from the level of cellular resolution but with respect to the assays used. I think part of the problem is the syncytial nature of the tissue, which makes it hard to analyze. Definitive proof will probably require live imaging in intact flies, something that I would thing feasible with 2 photon microscopy.

We provide new lineage tracing data based on the highly specific satellite cell

marker Zfh1 as well as G-trace techniques that shows both the lineal origin of the

satellite cells from AMP lineages and demonstrates that in response to injury the

labeled satellite cells proliferate and generate lineal progeny that fuse with

damaged muscle fibers. These new findings are presented in several new figures

and are described in new text.

Major issues:1) Are the muscle nuclei polyploid? Are the green arrow nuclei diploid? If so it would help to DAPI quantify.

Staining with DAPI is presented in multiple panels though out manuscript and

they are not polyploid. The cells indicated using green arrows are indeed diploid

in nature.

2) In Figure 1A there is only one superficial nuclei. In 1C there are three. The scale bar suggests the picture is the same size but 1C looks smaller?

We present new data in a new Figure 1 as well as in new subsequent figures,

which clearly show multiple superficial nuclei.

3) "(In control experiments, MARCM clones were also triggered in pupal stages and recovered in the adult to confirm that the labeled cells were not infiltrating cells that might derive from unknown proliferating cells located external to the wing disc; see Figure 3—figure supplement 1A-C)”. I don't understand this. Is the reasoning that during pupation there are no cell divisions in this lineage and therefore you would expect not to recover clones? Why is it parentheses? Also, throughout the paper there are examples of text placed in parentheses. Is this a convention the authors use to indicate an afterthought? It's unnecessary.

We have carried out new experiments and have removed this part of the text. With the new specific maker for satellite cells permitting the demarcation of the lineal origin of stem cells through pupal stages we now not only establish the lineage but also rule out that satellite cells might derive from other proliferating cells.

4) "Note that, while differentiated myoblasts are also targeted in this MARCM experiment, no GFP labeled cells are visible within the adult muscle fibres since their membrane tethered GFP becomes diffuse due to incorporation into the extensive muscle fibre membrane following cell fusion (see schematic in Figure 3D)." Wouldn't the argument used by the authors later, that gal80 is present, explain why there is no membrane GFP?

We have removed this figure and replaced it with new data and figures in the revised version. Moreover, we agree with reviewer on the fact that gal80 present in the muscle prevents labeling of fused cell nuclei in MARCM experiments. Indeed, this is further evidence for the unfused nature of labeled satellite cells.

5) "Care was taken to restrict damage such that only 1 or 2 muscle fibres were affected and that fibres were not severed by the injury (Figure 4A, B)." Out of how many fibres? How much area would this represent?

This is now much clearly presented multiple panels though out manuscript and it is always 1-2 out 6 DLM muscles. This is much clearer in low zoom images and this represents about one tenth of muscle area.

6) "DLMs damaged in this way can regenerate and in morphological respects appear normal after approximately 3 weeks (Figure 4—figure supplement 1)."

We now present new data on the nature of the injury and the timeline of regeneration at 2, 5 and 10 days in a new Results section and note that at 10 days the regeneration is largely complete.

7) How do they appear after 1 day? 2days? 3 days? etc. It's confusing why one would look after 3 weeks, especially when data presented later implies that muscle function returns after 2 days! By looking at many early time points one might get an idea of how the proposed repair occurs. In fact, maybe the area of damage is repaired by fusion of remaining tissue or hypertrophy of adjacent nuclei. There is no way to rule these alternatives out based on the resolution (spatial and temporally) provided. Also, it's unclear where the original injury might have been. There is no presentation of muscles that have no/poor repair (e.g. DmefGal4 X Notch RNAi) to compare with.

See response above to point 6).

8) "For both PH-3 and EdU labeling, evidence for increased mitotic activity was largely restricted to the damaged muscle fibres and rarely observed in undamaged muscle fibres”. Could you show a damaged fibre next an undamaged fibre with PH3 staining.

We have eliminated the corresponding figure in the revised version and now characterize satellite cell proliferation due to injury using the satellite cell specific Zhf1 label. Moreover, we also use G-trace to label the satellite cells after injury. Both new experiments show both damaged and undamaged fibres which support the above mentioned spatial distribution of proliferation.

9) In contrast, within the damaged muscle fibre, evidence for increased mitotic activity was seen in nuclei along the entire extent of the fibre length. It surprises me that mitosis would be seen along the entire fibre. How many nuclei would that be? Is that the case with vertebrate satellite muscle cells? How many of new progeny fuse with the fibre?

See response to point 8).

10) "In order to label only cells that were generated by mitotic activity in the adult, clones were induced in the adult 6h prior to physical injury and recovered 24h later." According to the authors divisions are rare, so how were clones recovered at all? The better way is follows right after in the parentheses.

The corresponding experiment has been removed in the revised version and replaced by Zfh1 based G-trace lineage tracing methods performed in the injured adult.

11) "This implies that some of the daughter cells generated by satellite cells during injury-induced proliferative activity remained at the muscle cell surface while others appear to have entered muscle fibre's interior and may have fused with the injured muscle cell." This was not convincing. They may have not entered the interior but rather intermixed in the space which has been disrupted by injury. Or that injury leads to non-specific fusion caused by injury.

The corresponding experiment has been removed in the revised version and replaced by Zfh1 based G-trace lineage tracing methods performed in the injured adult.

12) The FUCCI system does not provide information about lineage. And the quantity and quality of the data was not better than PH3. Along these lines, could the authors show examples of metaphase, anaphase and telophase? This would be much more convincing that the surprisingly numerous PH3 positive cells. Also, the long chain of "proliferative " cells shown using the FUCCI system was never seen or described in earlier experiments and makes me wonder if it's an artifact.

The corresponding experiment has been removed in the revised version and replaced by Zfh1 based G-trace lineage tracing methods performed in the injured adult.

13) Nts affects the entire animal. DmefGal4 X Notch RNAi affects all muscle. MARCM clones of Notch mutants would be more useful since one could see they are PH3 negative in a sea of PH3 positive WT cells.

Due to the highly restricted nature of the injury, localized effects can be clearly discerned and differentiated from possible global effects by the use of the new highly specific markers.

14) Was DAPT fed to the flies, for how long?

DAPT was fed for 3days after eclosion and injury was performed at the end of 48h and muscles were processed at the end of 72h.

15) The Notch responsive element is not a GFP construct of E(spl). It's, I believe, Su(H) binding sites fused to Gbe binding sites and based on the original reporter made by Sarah Bray's lab.

The corresponding experiment has been removed in the revised version.

16) "Targeted Notch downregulation led to significant perturbation in the flight initiation response, indicating the importance of satellite cell lineage proliferation and fusion to restore muscle function after injury." Actually, this is merely a strict correlation. That is, PH3 falls and so does flight initiation. But whether they are directly connected cannot be concluded. Notch may be required in Dmef2 positive cells for other things. I am also concerned and surprised that repair occurs in 2 days. And along these lines why in Figure 4—figure supplement 1 did the authors wait until 3 weeks for evidence of repair? Furthermore, what about Notch RNAi after 4 days, 5 days? (They only look at Notch RNAi after 2 days). Given the huge variation even among each genotype it's hard to see how one can conclude statistical significance. Can it recover? I would not expect it to given how well PH3 levels fall. Therefore, if it recovered it would argue something else is going on. And why only do 3 trials? Given the ease of the experiment you could do 300 trials.

The corresponding experiment has been removed in the revised version.

17) It's unclear why the authors chose a red box next to the area of damage as opposed to a box with the area of damage in the middle. And why a 100uM and not more? At what distance does PH3 fall? Or is it an entire fibre becomes activated?

The corresponding figure has been removed in the revised version.

Reviewer #2:The manuscript by Gunage et al. is a significantly improved version of a manuscript submitted earlier this year, that attempts to describe and characterize for the first time satellite cells in Drosophila. This appears a particularly difficult task, given a paucity of suitable markers for these cells and no established muscle injury model in this organism. Nevertheless, the significance of this work, if proven, would be high.The prior version of the manuscript suffered from a lack of suitable counterstains and marker analysis for the satellite cells, and this has been improved. It is clear that a separate population of cells exists on the flight muscle surface, and that upon injury there is an increase in the number of proliferating cells in and around the flight muscles. However, I still remain a little unconvinced that this revised work still definitively shows the origin, existence and activity of these cells.Firstly, there is still no good data supporting that the satellite cells arise from the wing imaginal discs, versus any other discs or any other part of the body that expresses Dmef2-gal4. The authors indicate they studied pupal time points in a supplementary figure, however that data is poorly described and does not appear to represent a time course.

We thank reviewer for these valuable and point specific comment. We now present extensive spatiotemporal data on satellite cell origin from larval through pupal stages which clearly show that satellite cells are lineal descendants of AMPs in the wing disc. This work is based on the newly identified Zfh1 marker.

Secondly, the authors report that they were not able to identify any markers of the satellite cells from a candidate list. It is unclear if they have tested Dmef2 itself. The satellite cells should be positive for this protein because the Dmef2-gal4 driver is active in the cells.

We thank the reviewer for this important comment. As a consequence, we have screened for such a marker and have discovered that Zfh-1 is a highly specific marker for satellite cells in the adult musculature. Data based on this maker are included throughout the extensively revised manuscript.

Thirdly, the authors are still struggling to definitively demonstrate that the satellite cells and their daughter cells correspond to the nuclei that are thought to infiltrate the injured muscle. In Figure 4D, for example, it looks as though essentially every nucleus becomes PH3 positive. Do the endogenous muscle nuclei (i.e. those that were present in the muscle prior to injury) also accumulate PH3? A clarification of not only the number of PH3-poistive nuclei in each sample, but also the total number of nuclei, would help resolve this. This point is not well supported by the data in Figure 4H-K, where the PH3 stain is so weak it could be argued to be either present in all nuclei or absent from all nuclei.In addition, the notion that some of the activated satellite cells remain on the surface of the muscle is attractive, but is not well supported by the existing data. Here, there is no proof that the surface nuclei are not fused to the muscle (and no support from the data in panel 4S-U), nor that the activated cells do not fuse to the muscle in the following day or so. Later time points following injury and MARCM induction might resolve this issue.

We have replaced these experiments by new Zfh1 driven G-trace experiments which clearly demonstrate that satellite cell lineal descendants fuse with muscle following injury.

Finally, it is still not clear what is the level of repair taking place. The rescue of flight defects is modest and not necessarily due to muscle defects (it could be some form of shock or inflammatory response that causes a reduction in flight after injury); the authors do not provide any data suggesting that new myofibrillar proteins are being made; and there is no indication that there is an overall increase in the number of muscle nuclei following repair. Thus, the roles of these cells are not defined.

We have eliminated the flight data in the revised manuscript. Moreover, we present a spatiotemporal characterization of the injury repair process. We feel that an in–depth characterization of the molecular mechanisms of the repair process goes beyond the scope of this manuscript which is focused on identification and characterization of satellite cells in normal and injured muscles.

[Editors' note: the author responses to third round of review follow.]

Reviewer #1[…] 1) Subsection “Two different types of cells are present in adult flight muscle”: "This observation is confirmed by co-staining these adult muscle fibres for expression of either Act88F, an indirect flight muscle specific isoform of actin, or Tropomoysin." Is this data contained in the manuscript? If so, it should be properly called out and if not, it should be added.

We are grateful to the reviewer for highlighting this gap. As you have suggested, we have added a supplement to Figure 1 (Figure 1—figure supplement 1) where we show unfused muscle associated mononucleate cells located at the surface of, and between, adult myofibres within whom A) DLM specific Actin (Act88F) and B) Tropomyosin localize to myofibrils. These stainings reiterate the localization of satellite cells illustrated in Figure 1.

2) The authors should provide in the Discussion section more information about their newly defined marker Zfh-1, including its other known functions in the fly (it appears to be involved in muscle development and in self-renewal of germline stem cells) as well as information about whether its homolog is expressed in mammalian satellite cells (for which multiple publicly available gene expression data sets are available).

We thank the reviewer for this important suggestion. Clearly it is essential to describe Zfh1 function related knowledge, from the perspective of this study in this manuscript. In the discussion, we have summarized in vivoinvestigations into the role of Zfh1 and its mammalian homolog ZEB in development. The summary can be found in the Discussion section of this manuscript. This is a valuable addition to this manuscript.

3) Statistical tests used should be identified in the figure legends and methods should describe statistical approaches and randomization (if used).

We appreciate this suggestion from our reviewer. Accordingly, descriptions of statistical analyses and parameters have been added to the figure legends to Figure 11D; Figure 12C and F; Figure 13C and E. A section on the statistics we employed has been added to the Materials and methods section as well.

4) If technically feasible, immunoEM to demonstrate that the "satellite cells" identified by ultrastructural analysis are the same cells as those identified by confocal microscopy and Zfh-1 expression would greatly enhance the authors' conclusions.

We agree with the reviewer on the value of demonstrating Zfh1 expression in satellite cell nuclei, delienating apposition to muscles, retaining their own cell membranes, with the resolution of electron microscopy. A clean immunoEM image of Zfh1 in satellite cells would robustly consolidate our observations made through immunohistochemistry. In the revision, we now readily acknowledge this in the Discussion section: “With improved immune-EM staining protocols on *Drosophila* DLMs, visualizing Zfh1 expression in these cells will prove valuable.” However, at this time, the best available protocols for immunoEM in DLMs (Reedy et al., 2000.) show heavy background. While we could try to optimize the protocols for our system and setup, all conceivable avenues of doing so require the generation of new reagents e.g. antibodies which would take months, and at a minimum a few more months for optimization of the protocol. Variability in quality of images is inherent to the combination of antibodies and EM for immunoEM in DLMs. The lack of guaranteed, unquestionably clean immunoEM images after the prospective effort and time, makes this experiment technically unfeasible at this time. This experiment will remain among our priorities for further studies, of which many are ongoing.